# Improving Decision Sparsity

**Yiyang Sun**
Duke University

**Tong Wang**
Yale University

**Cynthia Rudin**
Duke University

## Abstract

Sparsity is a central aspect of interpretability in machine learning. Typically, sparsity is measured in terms of the size of a model globally, such as the number of variables it uses. However, this notion of sparsity is not particularly relevant for decision making; someone subjected to a decision does not care about variables that do not contribute to the decision. In this work, we dramatically expand a notion of *decision sparsity* called the *Sparse Explanation Value* (SEV) so that its explanations are more meaningful. SEV considers movement along a hypercube towards a reference point. By allowing flexibility in that reference and by considering how distances along the hypercube translate to distances in feature space, we can derive sparser and more meaningful explanations for various types of function classes. We present cluster-based SEV and its variant tree-based SEV, introduce a method that improves credibility of explanations, and propose algorithms that optimize decision sparsity in machine learning models.

## 1 Introduction

The notion of *sparsity* is a major focus of interpretability in machine learning and statistical modeling [Tibshirani, 1996, Rudin et al., 2022]. Typically, sparsity is measured *globally*, such as the number of variables in a model, or as the number of leaves in a decision tree [Murdoch et al., 2019]. Global sparsity is relevant in many situations, but it is less relevant for individuals subject to the model's decisions. Individuals care less about, and often do not even have access to, the global model. For them, *local* sparsity, or **decision sparsity**, meaning the amount of information critical to *their own* decision, is more consequential.

An important notion of decision sparsity has been established in the work of Sun et al. [2024], which defined the Sparse Explanation Value (SEV), in the context of binary classification, as the number of factors that need to be changed to a reference feature value in order to change the decision. In contrast to SEV, counterfactual explanations tend not to be *sparse* since they require small changes to many variables in order to reach the decision boundary [Sun et al., 2024]. Instead, SEV provides sparse explanations: consider a loan application that is denied because the applicant has many delinquent payments. In that case, the decision sparsity (that is, the SEV) would be 1 because only a single factor was required to change the decision, overwhelming all possible mitigating factors. The framework of SEV thus allows us to see sparsity of models in a new light.

Prior to this work, SEV had one basic definition: it is the minimal number of features we need to set to their reference values to flip the sign of the prediction. The reference values are typically defined as the mean of the instances in the opposite class. This calculation is easy to understand, but somewhat limiting because the reference could be far in feature space from the point being explained and the explanation could land in a low density area where explanations are not credible. As an example, for the loan decision for a 21 year old applicant, SEV could create a counterfactual such as "Changing the applicant's 3-year credit history to 15 years would change the decision." While this counterfactual is valid, faithful, and sparse, it is *not close* because the distance between the query point and the counterfactual is so large (3 years to 15 years). In addition, this explanation is not *credible* because the proposed changes to the features lead to an unrealistic circumstance – 6-year-olds do not typically have credit. That is, the counterfactual does not represent a typical member of the opposite class.

38th Conference on Neural Information Processing Systems (NeurIPS 2024).

Lack of credibility is a common problem for many counterfactual explanations [Mothilal et al., 2020, Wachter et al., 2017, Laugel et al., 2017, Joshi et al., 2019]. Therefore, in this work, we propose to augment the SEV framework by adding two practical considerations, *closeness* of the reference point to the query, and *credibility* of the explanation, while also optimizing *decision sparsity*.

We propose three ways to create close, sparse and credible explanations. The first way is to create multiple possibilities for the reference, one at the center of each cluster of points (Section 4.1). Having a finite set of references keeps the references *auditable*, meaning that a domain expert can manually check the references prior to generating any explanations. By creating references spread throughout the opposite class, queries can be assigned to closer references than before. Second, we allow the references to be flexible, where their position can be shifted slightly from a central location in order to reduce the SEV (Section 4.4). The third way pertains to decision tree classifiers, where a reference point is placed on each opposite-class leaf, and an efficient shortest-path algorithm is used to find the nearest reference (Section 4.2). Table 1 shows a query at the top, and some SEV calculations from our methods below, showing feature values that were changed within the explanation.

Table 1: An example for a query in the FICO Dataset with different kinds of explanations, $SEV^1$ represents the SEV calculation with one single reference using population mean, $SEV^{©}$ represents the cluster-based SEV, $SEV^F$ represents the flexible-based SEV. $SEV^T$ represents the tree-based SEV The columns are four features.

| | EXTERNAL RISKESTIMATE | NUMSATIS-FACTORYTRADES | NETFRACTION REVOLVINGBURDEN | PERCENTTRADES NEVERDELQ | NUMFEATURE CHANGED |
|---|---|---|---|---|---|
| **Query** | 69.00 | 10.00 | 117.01 | 90 | |
| $SEV^1$ | **72.65** | **21.47** | **22.39** | 90 | 3 |
| $SEV^F$ | **78.00** | 10.00 | **9.00** | 90 | 2 |
| $SEV^{©}$ | **81.00** | **26.00** | **12.00** | 90 | 3 |
| $SEV^T$ | 69.00 | 10.00 | 117.01 | **100** | 1 |

In addition to developing methods for calculating SEV, we propose two algorithms to optimize a machine learning model to reduce the number of points that have high SEV without sacrificing predictive performance in Section 5, one based on gradient optimization, and the other based on search. The search algorithm is exact. It uses an exhaustive enumeration of the set of accurate models to find one with (provably) optimal SEV.

Our notions of decision sparsity are general and can be used for any model type, including neural networks and boosted decision trees. Decision sparsity can benefit any application where individuals are subject to decisions made from predictive models – these are cases where decision sparsity is more important than global sparsity from the individual perspectives.

## 2 Related Work

The concept of SEV revolves around finding models that are simple, in that the explanations for their predictions are sparse, while recognizing that different predictions can be simple in different ways (i.e., involving different features). In this way, it relates to (i) instance-wise explanations (iii) local sparsity optimization Models, which seek to explain and provide predictions of complex models. We further comment on these below.

**Instance-wise Explanations.** Prior work has developed methods to explain predictions of black boxes [e.g., Guidotti et al., 2018, Ribeiro et al., 2016, 2018, Lundberg and Lee, 2017, Baehrens et al., 2010] for individual instances. These explanations are designed to estimate importance of features, are not necessarily faithful to the model, and are not associated with sparsity in decisions, so they are fairly distant from the purpose of the present work. Our work is on tabular data; there is a multitude of unrelated work on explanations for images [e.g., Apicella et al., 2019, 2020] and text [e.g., Lei et al., 2016, Li et al., 2016, Treviso and Martins, 2020, Bastings et al., 2019, Yu et al., 2019, 2021]. More closely related are *counterfactual explanations*, also called inverse classification [e.g., Mothilal et al., 2020, Wachter et al., 2017, Lash et al., 2017, Sharma et al., 2024, Virgolin and Fracaros, 2023, Guidotti et al., 2019, Poyiadzi et al., 2020, Russell, 2019, Boreiko et al., 2022, Laugel et al., 2017, Pawelczyk et al., 2020]. Counterfactual explanations are typically designed to find the closest instance to a query point with the opposite prediction, without considering sparsity of

the explanation. However, extensive experiments [Delaney et al., 2023] indicate that these "closest counterfactuals" tend to be unnatural for humans because the decision boundary is typically in a region where humans have no intuition for why a point belongs to one class or the other. For SEV, on the other hand, reference values represent the population commons, so they are intuitive. Thus, SEV has two advantages over standard counterfactuals: its references are meaningful because they represent population commons, and its explanations are *sparse*.

**Local Sparsity Optimization Models**   While there are numerous prior works on developing post-hoc explanations, limited attention has been paid to developing models that provide sparse explanations. We are aware of only one work on this, namely the Explanation-based Optimization (ExpO) algorithm of Plumb et al. [2020] that used a neighborhood-fidelity regularizer to optimize the model to provide sparser post-hoc LIME explanations. Experiment in Appendix K in our paper shows that ExpO is both slower and provides less sparse predictions than our algorithms.

## 3   Preliminaries and Motivation

The Sparse Explanation Value (SEV) is defined to measure the sparsity of individual predictions of binary classifiers. The point we are creating an explanation for is called the *query*. The SEV is the smallest set of feature changes from the query to a reference that can flip the prediction of the model. When we make a change to the query's feature, we *align* it to be equal to that of the reference point. The reference point is a "commons," i.e., a prototypical point of the opposite class as the query. In this section, we will focus on the basic definition of SEV, the selection criteria for the references, as well as three reference selection methods.

### 3.1   Recap of Sparse Explanation Values

We define SEV following Sun et al. [2024]. For a specific binary classification dataset $\{\boldsymbol{x}_i, y_i\}_{i=1}^n$, with each $\boldsymbol{x}_i \in \mathbb{R}^p$, and the outcome of interest is $y_i \in \{0, 1\}$. (This can be extended to multi-class classification by providing counterfactuals for every other class than the query's class.) We predict the outcome using a classifier $f : \mathcal{X} \rightarrow \{0, 1\}$.

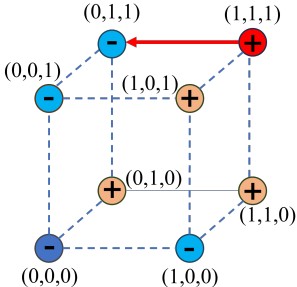

Figure 1: SEV Hypercube

Without loss of generality, in this paper, we are only interested in queries predicted as positive (class 1). We focus on providing a sparse explanation from the query to a *reference* that serves as a population commons, denoted $\boldsymbol{r}$. Human studies [Delaney et al., 2023] have shown that contrasting an instance with prototypical instances from another class provides more intuitive explanations than comparing it with instances from the same class. Thus, we define our references in the opposite class (negative class in this paper). To calculate SEV, we will align (i.e., equate) features from query $\boldsymbol{x}_i$ and reference $\tilde{\boldsymbol{x}}$ one at a time, checking at each time whether the prediction flipped. Thinking of these alignment steps as binary moves, it is convenient to represent the $2^p$ possible different alignment combinations as vertices on the boolean hypercube. The hypercube is defined below:

**Definition 3.1** (SEV hypercube). A SEV hypercube $\mathcal{L}_{f,i,\boldsymbol{r}}$ for a model $f$, an instance $\boldsymbol{x}_i$ with label $f(\boldsymbol{x}_i) = 1$, and a reference $\boldsymbol{r}$, is a graph with $2^p$ vertices. Here $p$ is the number of features in $\boldsymbol{x}_i$ and $\boldsymbol{b}_v \in \{0, 1\}^p$ is a Boolean vector that represents each vertex. Vertices $u$ and $v$ are adjacent when their Boolean vectors differ in one bit, $\|\boldsymbol{b}_u - \boldsymbol{b}_v\|_0 = 1$. 0's in $\boldsymbol{b}_v$ indicate the corresponding features are aligned, i.e., set to the feature values of the reference $\boldsymbol{r}$, while 1's indicate the true feature value of instance $i$. Thus, the actual feature values represented by the vertex $v$ is $\boldsymbol{x}_i^{\boldsymbol{r},v} := \boldsymbol{b}_v \odot \boldsymbol{x}_i + (\boldsymbol{1} - \boldsymbol{b}_v) \odot \boldsymbol{r}$, where $\odot$ is the Hadamard product. The score of vertex $v$ is $f(\boldsymbol{x}_i^{\boldsymbol{r},v})$, also denoted as $\mathcal{L}_{f,i,\boldsymbol{r}}(\boldsymbol{b}_v)$.

The SEV hypercube definition can also be extended from a hypercube to a Boolean lattice as they have the same geometric structure. There are two variants of the Sparse Explanation Value: one gradually aligns the query to the reference (SEV$^-$), and the other gradually aligns the reference to the query (SEV$^+$). In this paper, we focus on SEV$^-$:

Table 2: Calculation process for SEV$^-$ = 1

|  | TYPE | HOUSING | LOAN | EDUCATION | $Y$(RISK) |
|---|---|---|---|---|---|
| **(1,1,1)** | query | Rent | >10k | High School | High |
| **(0,1,1)** | SEV$^-$ Explanation | **Owning** | >10k | High School | **Low** |
| **(0,0,0)** | reference | Owning | <5k | Master | Low |

**Definition 3.2** (SEV$^-$). For a positively-predicted query $\boldsymbol{x}_i$ (i.e., $f(\boldsymbol{x}_i) = 1$), the Sparse Explanation Value Minus (SEV$^-$) is the minimum number of features in the query that must be aligned to reference $\boldsymbol{r}$ to elicit a negative prediction from $f$. It is the length of the shortest path along the hypercube to obtain a negative prediction,

$$\text{SEV}^-(f, \boldsymbol{x}_i, \boldsymbol{r}) := \min_{\boldsymbol{b} \in \{0,1\}^p} \quad \|\mathbf{1} - \boldsymbol{b}\|_0 \quad \text{s.t.} \quad \mathcal{L}_{f,i,\boldsymbol{r}}(\boldsymbol{b}) = 0.$$

Figure 1 and Table 2 shows an example of SEV$^-$=1 in a credit risk evaluation setting. Since $p = 3$, we construct a SEV hypercube with $2^3 = 8$ vertices. The red vertex $(1, 1, 1)$ corresponds to the query. The dark blue vertex at $(0, 0, 0)$ represents the negatively-predicted reference value. The orange vertices are predicted to be positive, and the light blue vertices are predicted to be negative. To compute SEV$^-$, we start at $(1, 1, 1)$ and find the shortest path to a negatively-predicted vertex. On this hypercube, $(0, 1, 1)$ is closest. Translating this to feature space, this means that if the query's housing situation changes from renting to the reference value "owning," it would be predicted as negative. This means that **SEV$^-$ is equal to 1** in this case. The feature vector corresponding to this closest vertex $(0, 1, 1)$, is called the **SEV$^-$ explanation** for the query, denoted by $\boldsymbol{x}_i^{\boldsymbol{r},\text{expl}}$ for reference $\boldsymbol{r}$.

## 3.2 Motivation of Our Work: Sensitivity to Reference Points

Since SEV$^-$ is determined by the path on a SEV hypercube and each hypercube is determined by the reference point, the SEV$^-$ is therefore sensitive to the selection of reference points. Adjusting the reference point trades off between *sparsity* (according to SEV$^-$) and *closeness* (measured by $\ell_2$, $\ell_\infty$ (see Section 6.1) or $\ell_0$ (see Section 6.4) distance between the query and its assigned reference point). Note that this trade-off exists because SEV$^-$ tends to be small when the reference is far from the query. More detailed explanations, visualizations, and experiments are shown in Appendix B.

**Selecting References.** The reference must represent the commons, meaning the negative population, and the generated explanations should represents the negative populations as well. Moreover, the negative population may have subpopulations; e.g., Diabetes patients may have higher blood glucose levels, while hypertension patients have higher blood pressure. To have meaningful coverage of the negative population, in this work, we consider *multiple* references, placed *within the various subpopulations*. This allows each point in the positive population to be closer to a reference. Let $\mathcal{R}$ denote possible placements of references. For query $\boldsymbol{x}_i$, an individual-specific reference $\boldsymbol{r}_i \in \mathcal{R}$ for $\boldsymbol{x}_i$ is chosen based on three criteria: it should be nearby (i.e., close), and should provide a sparse and reasonable explanation. That is, we are looking to minimize the following three objectives over placement of the reference $\boldsymbol{r}_i$:

$$\|\boldsymbol{x}_i - \boldsymbol{r}_i\|, \boldsymbol{r}_i \in \mathcal{R} \quad \text{(Closeness)} \tag{1}$$

$$\text{SEV}^-(f, \boldsymbol{x}_i, \boldsymbol{r}_i), \boldsymbol{r}_i \in \mathcal{R} \quad \text{(Sparsity)} \tag{2}$$

$$-P(\boldsymbol{x}_i^{\text{expl},\boldsymbol{r}_i} | X^-) \quad \text{(Negated Credibility)}, \tag{3}$$

with the constraint that the references obey auditability, meaning that domain experts are able to check the references manually, or construct them manually. The function $\text{SEV}^-(f, \boldsymbol{x}_i, \boldsymbol{r}_i)$ in (2) represents the SEV$^-$ computed with the given function $f$, query $\boldsymbol{x}_i$, and the individual-specific reference $\boldsymbol{r}_i$ for generating the hypercube. $\boldsymbol{x}_i^{\text{expl},\boldsymbol{r}_i}$ is the sparse explanation for the sample $\boldsymbol{x}_i$, and $P(\cdot | X^-)$ in the definition of credibility represents the probability density distribution of the negative population and $P(\boldsymbol{x}_i^{\text{expl},\boldsymbol{r}_i} | X^-)$ is the density of the negative distribution at $\boldsymbol{x}_i^{\text{expl},\boldsymbol{r}_i}$. If $P(\boldsymbol{x}_i^{\text{expl},\boldsymbol{r}_i} | X^-)$ is large, $\boldsymbol{x}_i^{\text{expl},\boldsymbol{r}_i}$ is in a high-density region.

## 4 Meaningful and Credible SEV

We now describe cluster-based SEV, which improves closeness at the expense of SEV, and its variant, tree-based SEV, which improves all three objectives and computational efficiency. We also present methods to improve the credibility and sparsity of the explanations.

### 4.1 Cluster-based SEV: Improving Closeness

This approach creates multiple references for the negative population. A clustering algorithm is used to group negative samples, and the resulting cluster centroids are assigned as references. A query is assigned to its closest cluster center:

$$\tilde{\boldsymbol{r}}_i \in \arg\min_{\boldsymbol{r}\in\mathcal{C}} \|\boldsymbol{x}_i - \boldsymbol{r}\|_2$$

where $\mathcal{C}$ is the collection of centroids obtained by clustering the negative samples. We refer to the $SEV^-$ produced by the grouped samples as cluster-based SEV, denoted $SEV^{\copyright}$. Figure 2 illustrates the calculation of $SEV^{\copyright}$ for two examples located in two different centroids. A red dot represents a query, while a blue dot represents a reference.

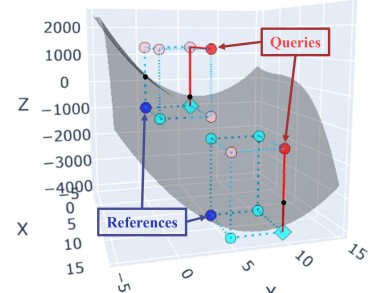

Figure 2: Cluster-based SEV

For each instance, it selects the closest centroid and considers the SEV hypercube, where each cyan point represents a negatively predicted vertex and each pink point represents a positively predicted vertex. We deduce by following the red lines that the $SEV^{\copyright}$ for the two queries are 2 and 1, respectively. The cluster centroids should serve as a cover for the negative class. To ensure that the cluster centroids have negative predictions, we use the soft clustering method of Bezdek et al. [1984] to constrain the predictions of the cluster centers. Details are in Appendix C.

## 4.2 Tree-based SEV: $SEV^{\copyright}$ Variant with Useful Properties and Computational Benefits

Tree-based SEV is a special case of cluster-based SEV, where we consider each negative leaf as a reference candidate, and find the sparsest explanation (path along the tree) to the nearest reference. Here, $SEV^-$ and $\ell_0$ distance (i.e., edit distance) are equivalent. That is, we find the minimum number of features to change in order to achieve a negative prediction.

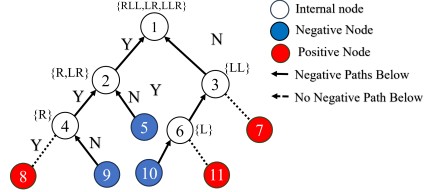

Figure 3: $SEV^T$ Preprocessing

We denote $SEV^T$ as the $SEV^-$ calculated based on this process. Here, we assume that trees have no trivial splits where all child leaves make the same prediction. If so, we would collapse those leaves before calculating the $SEV^T$. The first theorem below refers to decision paths that have negatively predicted child leaves. This is where taking one different choice at an internal split leads to a negative leaf.

**Theorem 4.1.** *With a single decision classifier DT and a positively-predicted query $\boldsymbol{x}_i$, define $N_i$ as the leaf that captures it. If $N_i$ has a sibling leaf, or any internal node in its decision path has a negatively-predicted child leaf, then $SEV^T$ is* **equal to 1**.

The second theorem states that $SEV^-$ and minimum edit distance from the query to negative leaves are equivalent.

**Theorem 4.2.** *With a single decision tree classifier DT and a positively-predicted query $\boldsymbol{x}_i$, with the set of all negatively predicted leaves as reference points, both $SEV^-$ and the $\ell_0$ distance (edit distance) between the query and the $SEV^-$ explanation are minimized.*

The proofs of those two theorems are shown in Appendix L and M. The structure of tree models yields an extremely efficient way to calculate $SEV^-$. We perform an important preprocessing step before any $SEV^-$

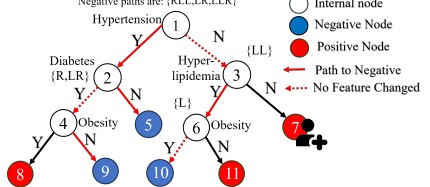

Figure 4: Efficient $SEV^T$ calculation: Query (node ⑦) has $SEV^T$=1, which goes to node ⑩. The path to this node is recorded as LL at node ③, which is along the decision path to node ⑦.

calculations are done, which will make $SEV^-$ easier to calculate for all queries at runtime. At each internal node, we record all paths to negative leaves anywhere below it in the tree. This is described in Algorithm 2 in Appendix E. E.g., if the tree has binary splits, a path from an internal node to a leaf node might require us to go left, then right, then left. In that case, we store LRL on this internal node to record this path. Then, when a query arrives at runtime (in a positive leaf, since it has a positive prediction), we traverse directly up its decision path all the way to the root node. For all internal nodes in the decision path, we observe distances to each negative leaf, which were stored during

preprocessing. We traverse each of these, and the minimum distance among these is the $\text{SEV}^-$. This is described in Algorithm 3 in Appendix E and illustrated in Figure 4. Note that we actually would traverse to each negative node because some internal decisions might not need to be changed along the path. In the example in Figure 4, we change the split at node ③, and use the value that the query already has for the split at node ⑥, landing in node ⑩, so $\text{SEV}^-$ is 1 not 2.

Table 3: Illustration of $\text{SEV}^T$ calculation.

| | ACTION | HYPER-TENSION | DIABETES | HYPER-LIPIDEMIA | OBESITY | HAVE STROKE | # OF CHANGED CONDITION (SEV) |
|---|---|---|---|---|---|---|---|
| **Instance** ①→③→⑦ | **Check node** ①&③ | No | Yes | No | Yes | Yes ⑦ | |
| **Flip at node** ③ | **Check LL** | No | | Yes | Yes | No ⑩ | 1 |
| | ③→⑥→⑩ | | | Flip at ③ | (Unchanged) | | |
| **Flip at node** ① | **Check LR** | Yes | No | | | No ⑤ | 2 |
| | ②→⑤ | Flip at ① | Flip at ② | | | | |
| | **Check LLR** | Yes | Yes | | No | No ⑨ | 2 |
| | ②→④→⑨ | Flip at ① | (Unchanged) | | Flip at ④ | | |

Table 3 walks through the calculation again, using the names of the features (hypertension, diabetes, etc.). On the first action line, the decision path to the query is ③→⑥→⑩. That means we check ① and ③ for negative paths, yielding path LL. We flip node ③ (change Hyperlipidemia to 'yes') and follow the LL path. We do not change Obesity to get to the negative node, so we record the $\text{SEV}^T$ as 1 in that row. In our implementation, we simply stop when we reach an $\text{SEV}^T$=1 solution, but we will continue in order to illustrate how the calculation works. We go up to node ① and repeat the process for the LR and LLR paths. Those both have $\text{SEV}^T$=2.

Note that the reference can be any point $x$ within the leaf; if the leaf is defined by thresholds such as $3 < x_1 < 5$ and $x_2 > 7$, then any point satisfying those conditions is a viable reference. Given a query, the algorithm flips some of its feature values to satisfy conditions of a leaf with the opposite prediction. Since any point in the leaf is a viable reference, we could choose the median/mean values of points in the leaf as the references, or a more meaningful value. That choice will not influence the fast calculation of SEV-T.

## 4.3 Improving Credibility for All SEV Calculations

As we mentioned in Section 3.2, the credibility objective encourages explanations to be located in high-density region of the negative population. Previous $\text{SEV}^-$ definitions focus on sparsity and closeness objectives, but did not consider credibility. It is possible to increase credibility easily while constructing an explanation: if the explanation veers out of the high-density region, we continue walking along the SEV hypercube during SEV calculations. Specifically, we continue moving towards the reference until the vertex is in a high-density region. Since the reference is in a high-density region, walking towards it will eventually lead to a high-density point. The tree-based SEV explanations automatically satisfy high credibility:

**Theorem 4.3.** *With a single sparse decision tree classifier $DT$ with support at least $S$ in each negative leaf, the $\text{SEV}^T$ explanation for query $x_i$ always satisfies credibility at least $\frac{S}{N^-}$, where $N^-$ is the total number of negative samples.*

This theorem can be easily proved because $\text{SEV}^-$ explanations generated by $\text{SEV}^T$ are always the negative leaf nodes (which are the references), and the references are located in regions with support at least $S$ by assumption.

## 4.4 Flexible Reference SEV: Improving Sparsity

From Section 3.2, we know that queries further from the decision boundary tend to have lower $\text{SEV}^-$. Based on this, we introduce Flexible Reference SEV (denoted $\text{SEV}^F$), which moves the

reference value slightly in order to achieve a lower value of the model output $f(\tilde{r})$ given a reference $\tilde{r}$, and the decision function for classification $f(\cdot)$, which, in turn, is likely to lead to lower $\text{SEV}^-$. The optimization for finding the optimal reference is: $r^* \in \arg\min_r f(r)$   s.t $\|r - \tilde{r}\|_\infty \le \epsilon_F$ where the $\arg\min$ is over reference candidates that are near the original reference value $\tilde{r}$. The flexibility threshold $\epsilon_F$ represents the flexibility allowed for moving the reference within a ball. We limit flexibility so the explanation stays meaningful. Since it is impractical to explore all potential combinations of feature-value candidates, we address this problem by marginalizing. Specifically, we optimize the reference over each feature independently. The detailed algorithm for calculating Flexible Reference SEV, denoted $\text{SEV}^F$, is shown in Algorithm 1 in Appendix D. In Section 6.2, we show that moving the reference slightly can sometimes reduce the SEV, improving sparsity.

## 5   Optimizing Models for $\text{SEV}^-$

Above, we showed how to calculate $\text{SEV}^-$ for a fixed model. In this section, we describe how to train classifiers that optimize the average $\text{SEV}^-$ without loss in predictive performance. We propose two methods: minimizing an easy-to-optimize surrogate objective (Section 5.1) and searching for models with the smallest SEV from a "Rashomon set" of equally-good models (Section 5.2). In what follows, we assume that $\text{SEV}^-$ was calculated prior to optimization, that reference points were assigned to each query, and that these assignments do not change throughout the calculation.

### 5.1   Gradient-based SEV Optimization

Since we want to minimize expected test $\text{SEV}^-$, the most obvious approach would be to choose our model $f$ to minimize average training $\text{SEV}^-$. However, since SEV calculations are not differentiable and they are combinatorial in the number of features and data points, this would be intractable. Following Sun et al. [2024], we instead design the optimization objective to penalize each sample where $\text{SEV}^-$ is more than 1. Thus, we propose the loss term:

$$\ell_{\text{SEV\_All\_Opt}-}(f) := \frac{1}{n^+} \sum_{i=1}^{n^+} \max\left(\min_{j=1,\dots,p} f((\mathbf{1} - e_j) \odot x_i + e_j \odot \tilde{r}_i),\, 0.5\right),$$

where $e_j$ is the vector with a 1 in the $j^{th}$ coordinate and 0's elsewhere, $n^+$ is the number of queries, and the reference point $\tilde{r}_i$ is specific to query $x_i$ and chosen beforehand. Intuitively, $f((\mathbf{1} - e_j) \odot x_i + e_j \odot \tilde{r}_i)$ is the function value of query $x_i$ where its feature $j$ has been replaced with the reference's feature $j$. $\min_{j=1,\dots,p} f((\mathbf{1} - e_j) \odot x_i + e_j \odot \tilde{r}_i)$ chooses the variable to replace that most reduces the function value. If the $\text{SEV}^-$ is 1, then when this replacement is made, the point now is on the negative side of the decision boundary and $f$ is less than 0.5, in which case the $\max$ chooses 0.5. If $\text{SEV}^-$ is more than 1, then after replacement, $f$ will still predict positive and be more than 0.5, in which case, its value will contribute to the loss. This loss is differentiable with respect to model parameters except at the "corners" and not difficult to optimize.

To put these into an algorithm, we optimize a linear combination of different loss terms,

$$\min_{f \in \mathcal{F}} \ell_{\text{BCE}}(f) + C_1 \ell_{\text{SEV\_All\_Opt}-}(f) \tag{4}$$

where $\ell_{\text{BCE}}$ is the Binary Cross Entropy Loss to control the accuracy of the training model and $\mathcal{F}$ is a class of classification models that estimate the probability of belonging to the positive class. $\ell_{\text{SEV\_All\_Opt}-}$ is the loss term that we have just introduced above. $C_1$ can be chosen using cross-validation. We define **All-Opt$^-$** as the method that optimizes (4). Our experiments show that this method is not only effective in shrinking the average $\text{SEV}^-$ but often attains the minimum possible $\text{SEV}^-$ value of 1 for most or all queries.

### 5.2   Search-based SEV Optimization

As defined in Section 4.2, our goal is to find a model with the lowest average $\text{SEV}^-$ among classification models with the best performance.

The Rashomon set [Semenova et al., 2022, Fisher et al., 2019] is defined as the set of all models from a given class with performance approximately that of the best-performing model. The first method

that stores the entire Rashomon set of any nontrivial function class is called TreeFARMS [Xin et al., 2022], which stores all good sparse decision trees in a data structure. TreeFARMS allows us to optimize multiple objectives over the space of sparse trees easily by enumeration of the Rashomon set to find all accurate models, and a loop through the Rashomon set to optimize secondary objectives. We use TreeFARMS and search through the Rashomon set for a model with the lowest average $\text{SEV}^-$:

$$\min_{f \in \mathcal{R}_{\text{set}}} \frac{1}{n^+} \sum_{i=1}^{n^+} \text{SEV}^T(f, \boldsymbol{x}_i),$$

where the Rashomon set is $\mathcal{R}_{\text{set}}$, and where we use $\text{SEV}^T$ as the $\text{SEV}^-$ for each sparse tree in the Rashomon set. Recall that Algorithms 2 and 3 show how to calculate $\text{SEV}^T$. We call this search-based optimization as **TOpt**.

## 6 Experiments

**Training Datasets**   To evaluate whether our proposed methods would achieve sparser, more credible and closer explanations, we present experiments on seven datasets: (i) UCI Adult Income dataset for predicting income levels [Dua and Graff, 2017], (ii) FICO Home Equity Line of Credit Dataset for assessing credit risk, used for the Explainable Machine Learning Challenge [FICO, 2018], (iii) UCI German Credit dataset for determining creditworthiness [Dua and Graff, 2017], (iv) MIMIC-III dataset for predicting patient outcomes in intensive care units [Johnson et al., 2016a,b], (v) COMPAS dataset [Jeff Larson and Angwin, 2016, Wang et al., 2022a] for predicting recidivism, (vi) Diabetes dataset [Strack et al., 2014] for predicting whether patients will be re-admitted within two years, and (vii) Headline dataset for predicting whether the headline is likely to be shared by readers [Chen et al., 2023]. Additional details on data and preprocessing are in Appendix A.

**Training Models**   For $\text{SEV}^{\copyright}$, we trained four baseline binary classifiers: (i, ii) logistic regression classifiers with $\ell_1$ (L1LR) and $\ell_2$ (L2LR) penalties, (iii) a gradient boosting decision tree classifier (GBDT), and (iv) a 2-layer multi-layer perceptron (MLP), and tested its performance with $\text{SEV}^F$ added, and the credibility rules added. In addition, we trained All-Opt$^-$ variants of these models in which the SEV penalties described in the previous sections are implemented. For $\text{SEV}^T$ methods, we compared tree-based models from CART, C4.5, and GOSDT [Lin et al., 2020, McTavish et al., 2022] with the TOpt method proposed in Section 5.2. Details on training the methods is in Appendix F.

**Evaluation Metrics**   To evaluate whether good references are selected for the queries, we evaluate sparsity and closeness (i.e., similarity of query to reference). For **sparsity**, we use the average number of feature changes (which is the same as $\ell_0$ norm) between the query and the explanation; for **closeness**, we use the median $\ell_\infty$ norm between the generated explanation and the original query as the metric for $\text{SEV}^{\copyright}$. For tree-based models, we use only $\text{SEV}^T$ as the metric since $\text{SEV}^T$ and $\ell_0$ norm are equivalent; for **credibility**, we need some way of estimating $P(\cdot|X)$ since we cannot observe it directly, so we trained a Gaussian mixture model on the negative samples of each dataset, and used the mean log-likelihood of the generated explanations as the metric for $\text{SEV}^{\copyright}$ and $\text{SEV}^F$, for TOpt, since it has already been a sparse decision tree, then we don't need to calculate the credibility.

### 6.1 Cluster-based SEV shows improvement in credibility and closeness

Let us show that $\text{SEV}^{\copyright}$ provides improved explanations. Here, we calculated the metric for different $\text{SEV}^{\copyright}$ variants, $\text{SEV}^{\copyright}$ and $\text{SEV}^{\copyright+F}$($\text{SEV}^{\copyright}$ with flexible reference), and compared to the original $\text{SEV}^1$, where $\text{SEV}^1$ is defined as the $\text{SEV}^-$ calculation with single reference generated by the mean value of each numerical feature and mode value of each categorical feature of the negative population, as done in the original SEV paper [Sun et al., 2024] under various datasets and models.

Figure 5a shows the relationship between spasity and variants, the scatter plot between mean $\text{SEV}^-$ and mean $\ell_\infty$ for each explanation generated by different variants. We find that **$\text{SEV}^{\copyright}$ improves closeness**, which was expected since the references were designed to be closer to the queries. Interestingly, $\text{SEV}^{\copyright}$ sometimes has lower decision sparsity than $\text{SEV}^1$. $\text{SEV}^{\copyright}$ was designed to trade off $\text{SEV}^-$ for closeness, so it is surprising that it sometimes performs strictly better on both metrics, particularly for the COMPAS, Diabetes, and German Credit datasets.

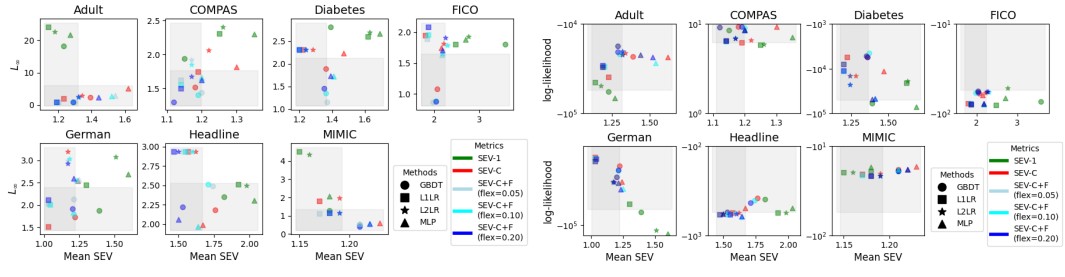

(a) Sparsity (SEV$^-$) and Closeness (L$_\infty$)     (b) Sparsity (SEV$^-$) and Credibility (log-likelihood)

Figure 5: Explanation performance under different models and metrics. We desire lower SEV$^-$ for sparsity, lower $\ell_\infty$ for closeness and higher log likelihood for credibility (shaded regions)

Interestingly, we also find that even though we do not optimize credibility for our model, Figure 5b shows that SEV$^©$ improves credibility, particularly for the Adult, German, and Diabetes datasets by plotting the relationship between mean SEV$^-$ and mean log-likelihood of the generated explanations. It is reasonable since the references are the cluster centroids for the negative samples, so the explanations are more likely to be located in the same high-density area. More detailed values for those methods and metrics are shown in Appendix H.

## 6.2   Flexible Reference SEV can improve sparsity without losing credibility

In Section 4.4, we proposed the flexible reference method for sparsifying SEV$^-$ explanations, which moves the reference slightly away from the decision boundary. The blue points in Figure 5a and 5b have already shown that with small modification of the reference, the credibility of the explanations is not affected. Figure 6a shows how SEV$^-$ and credibility change as we increase flexibility; SEV$^-$ sometimes substantially decreases while credibility is maintained.

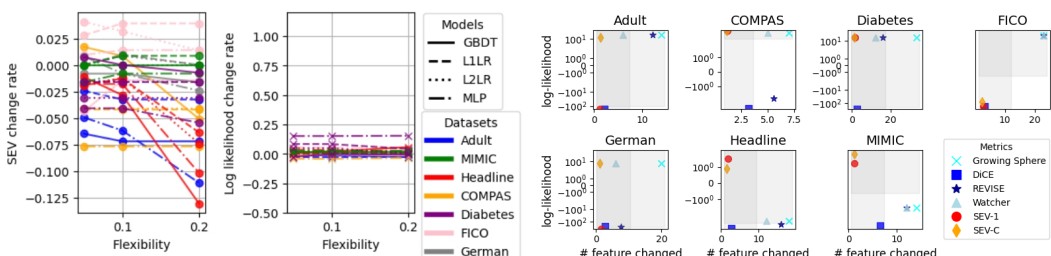

(a) SEV$^-$/Credibility change rate for varying flexibility   (b) Median Log likelihood and # of features changed

Figure 6: (a) Sparsity and Credibility as a function of the change of flexibility level (0 to 5%/10%/20%) under different models and datasets (b) The median log-likelihood and number of features within different counterfactual explanations. Points at the upper left corner are desired.

## 6.3   SEV$^-$ provides the sparsest explanation compared to other counterfactual explanations

Recall that SEV$^-$ flips features of the query to values of the population commons. This can be viewed as a type of counterfactual explanation, though typically, counterfactual explanations aim to find the minimal distance from one class to another. In this experiment, we compare the sparsity of SEV$^-$ calculations to that of baseline methods from the literature on counterfactual explanations, namely Watcher [Wachter et al., 2017], REVISE [Joshi et al., 2019], Growing Sphere [Laugel et al., 2017], and DiCE [Mothilal et al., 2020].

Figure 6b shows sparsity and credibility performance of all counterfactual explanation methods on different datasets under $\ell_2$ logistic regression (other information, including $\ell_\infty$ norms for counterfactual explanation methods, is in Appendix G). All SEV variants are in warm colors, while competitors are in cool colors. SEV$^-$ methods have the sparsest explanations, followed by DiCE. (A comparison of SEV$^-$ to DiCE is provided by Sun et al. [2024].) We point out that this comparison was made on

methods that were not designed to optimize explanation sparsity. Importantly, sparsity is essential for human understanding [Rudin et al., 2022]. Moreover, it has been shown that SEV (especially SEV$^{©}$) would have more credible explanations than competitors, while explanations remain sparse.

### 6.4 All-Opt$^{-}$ and TOpt optimize SEV$^{-}$, preserving model performance, explanation closeness and credibility

Even without optimization, our SEV$^{-}$ variants improve decision sparsity and/or closeness. If we are willing to retrain the prediction model as discussed in Section 5, we can improve these metrics further, creating accurate models with higher decision sparsity. Figure 7a shows that gradient-based SEV optimization can reduce the SEV without harming the closeness metric ($\ell_\infty$) and the credibility metrics (log-likelihood). The slashed bars represents the SEV$^{-}$, $\ell_\infty$ and log likelihood metrics before optimization using different models, while the colored bars are the results after optimizing with All-Opt$^{-}$. We have also compared our results with ExpO [Plumb et al., 2020], which is a optimization method that maximizes the mean neighborhood fidelity of the queries, but we have found that explanations are not sparse, and it requires long training times; the detailed results are shown in Appendix K.

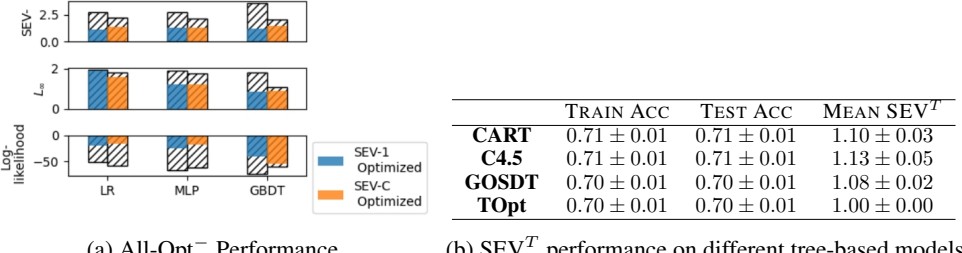

| | Train Acc | Test Acc | Mean SEV$^{T}$ |
|---|---|---|---|
| **CART** | $0.71 \pm 0.01$ | $0.71 \pm 0.01$ | $1.10 \pm 0.03$ |
| **C4.5** | $0.71 \pm 0.01$ | $0.71 \pm 0.01$ | $1.13 \pm 0.05$ |
| **GOSDT** | $0.70 \pm 0.01$ | $0.70 \pm 0.01$ | $1.08 \pm 0.02$ |
| **TOpt** | $0.70 \pm 0.01$ | $0.70 \pm 0.01$ | $1.00 \pm 0.00$ |

(a) All-Opt$^{-}$ Performance      (b) SEV$^{T}$ performance on different tree-based models

Figure 7: (a) SEV$^{-}$ and $\ell_\infty$ before and after All-Opt$^{-}$ on the FICO Dataset. Slashed bars are before, solid color is after. (b) All tree-based models with similar accuracy have low SEV$^{T}$.

For the Tree-based SEV, we have applied the efficient computation procedure to different kinds of tree-based models, and compared them with the search-based optimization method (TOpt) for trees in Section 5. The search-based algorithm works perfectly in finding a good model without performance loss. It achieves a perfect average SEV score of 1.00.

## Conclusion

Decision sparsity can be more useful than global model sparsity for individuals, as individuals care less about, and often do not even have access to, the global model. We presented approaches to achieving high decision sparsity, closeness and credibility, while being faithful to the model. One limitation of our method is that causal relationships may exist among features, invalidating certain transitions across the SEV hypercube. This can be addressed by searching across vertices that do not satisfy the causal relationship, though it requires knowledge of the causal graph. Another limitation is that to make the explanation more credible, the threshold to stop searching the SEV hypercube is not easy to determine. Future studies could focus on on these topics. Overall, our work has the potential to enhance a wide range of applications, including but not limited to loan approvals and employment hiring processes. Improved SEV translates directly into explanations that simply make more sense to those subjected to the decisions of models.

## Acknowledgement

We acknowledge funding from the National Science Foundation under grants DGE-2022040 and CMMI-2323978.

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

# A    Data Description and Preprocessing

The datasets were divided into training and test sets using an 80-20 stratification. The numerical features were transformed by standardization to have a mean of zero and a variance of one. The categorical features, which have $k$ different levels, were transformed into $k-1$ binary variables using one-hot encoding. The binary characteristics were transformed into a single dummy variable using one-hot encoding. The sizes of the datasets before and after encoding are shown in Table 4.

|  | OBSERVATIONS | PRE-ENCODED FEATURES | POST-ENCODED FEATURES |
|---|---|---|---|
| COMPAS | 6,907 | 7 | 7 |
| Adult | 32,561 | 14 | 107 |
| MIMIC-III | 48,786 | 14 | 14 |
| Diabetes | 101,766 | 33 | 101 |
| German Credit | 1,000 | 20 | 59 |
| FICO | 10,459 | 23 | 23 |
| Headlines | 41,752 | 12 | 17 |

Table 4: Training Dataset Sizes

Below we provide more details for each dataset.

**COMPAS**

The COMPAS dataset contains information on criminal recidivism in Broward County, Florida [Jeff Larson and Angwin, 2016]. The goal of this dataset is to predict the likelihood of recidivism within a two-year period, taking into account the following variables: gender, age, prior convictions, number of juvenile felonies/misdemeanors, and whether the current charge is a felony.

**Adult**

The Adult data is derived from U.S. Census statistics, including information on demographics, education, employment, marital status, and financial gain/loss [Dua and Graff, 2017]. The target variable of this dataset is whether an individual's salary exceeds $50,000.

**MIMIC-III**

MIMIC-III is a comprehensive database that stores a variety of medical data related to the experience of patients in the Intensive Care Unit (ICU) at Beth Israel Deaconess Medical Center [Johnson et al., 2016a,b]. The outcome of interest is determined by the binary indicator known as the "hospital expires flag," which indicates whether or not a patient died during their hospitalization. We chose the following set of variables as features: `age`, `preiculos` (pre-ICU length of stay), `gcs` (Glasgow Coma Scale), `heartrate_min`, `heartrate_max`, `meanbp_min` (min blood pressure), `meanbp_max` (max blood pressure), `resprate_min`, `resprate_max`, `tempc_min`, `tempc_max`, `urineoutput`, `mechvent` (whether the patient is on mechanical ventilation), and `electivesurgery` (whether the patient had elective surgery).

**Diabetes**

The Diabetes dataset is derived from 10 years (1999-2008) of clinical care at 130 hospitals and integrated delivery networks in the United States [Dua and Graff, 2017]. It consists of more than 50 characteristics that describe patient and hospital outcomes. The dataset includes variables such as `race`, `gender`, `age`, `admission type`, `time spent in hospital`, `specialty of admitting physician`, `number of lab tests performed`, `number of medications`, and so on. We consider whether the patient will return to the hospital within 2 years as a binary indicator.

**German Credit**

The German credit data [Dua and Graff, 2017] uses financial and demographic indicators such as checking account status, credit history, employment/marital status, etc., to predict whether an individual will default on a loan.

**FICO**

The FICO Home Equity Line of Credit (HELOC) dataset [FICO, 2018] is used for the Explainable Machine Learning Challenge. It includes a number of financial indicators, such as the number of inquiries on a user's account, the maximum delinquency, and the number of satisfactory transactions, among others. These indicators relate to different individuals who have applied for credit. The target variable is whether a consumer has been 90 or more days delinquent at any time within a 2-year period since opening their account.

**Headlines**

The News Headline dataset [Zhong et al., 2024] is a survey data aimed at discovering what kind of news content is shared and what factors are significantly associated with news sharing. The survey includes several factors, including, `age`, `income`, `gender`, `ethnicity`, `social protection`, `economic protection`, `truth` ("What is the likelihood that the above headline is true?"), `familiarity` ("Are you familiar with the above headline (have you seen or heard about it before)? )"), `Importance` ("Assuming the headline is completely accurate, how important would you consider this news to be?"), `Political Concordance` ("Assuming the above headline is completely accurate, how favorable would you consider it to be for Democrats versus Republicans?"). The goal of this data set is to predict `Sharing` ("If you were to see the above article on social media, how likely would you be to share it?").

# B    Sensitivity of the reference points

In this section, we will mainly show how sensitive SEV$^-$ is when we change the reference. Figure 8 shows an example of this, where moving the reference further away from the query (from $r$ to the $r'$) changes the SEV$^-$ from 2 to 1. In this figure, the dark blue axes represent the feature values of different reference values, while the black dashed line represents the decision boundary of a linear classifier. Areas with different colors represent data points with different SEV$^-$. When the reference moves further from the decision boundary (from $r$ to $r'$), the corresponding areas for SEV$^-$ will move away from the decision boundary. For example, the star located in the yellow area has an SEV$^-$ of 1 instead of 2 when the reference moves from $r$ to $r'$. If the reference point is $r$, then the query needs to align the feature values along both x and y-axis to reach the SEV Explanation with reference $r$ (recall an example of SEV$^-$ explanation in Figure 2) in Section 3.2, which is the same point as $r$. However, if the reference point is $r'$, then the query only needs to align the feature value along the x-axis to reach the SEV Explanation with SEV= 1, which is the light blue dot.

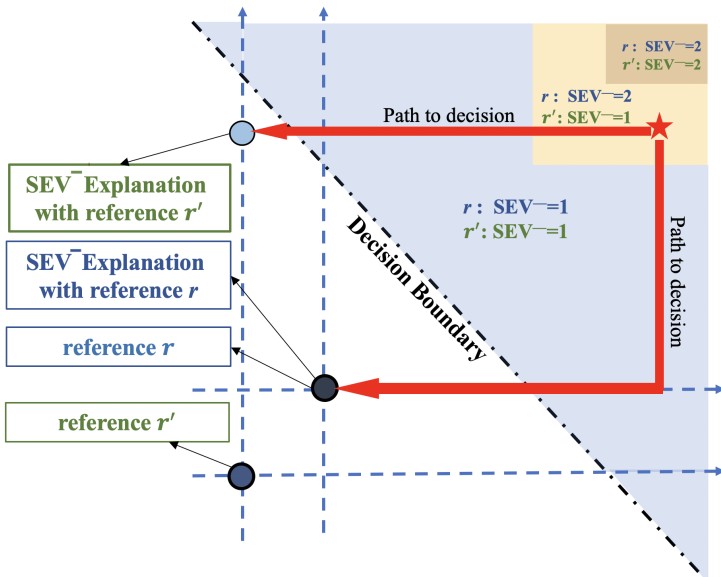

Figure 8: SEV$^-$ distribution

Experiments have also shown that moving data points closer to the decision boundary might increase SEV$^-$. The result on the Explainable ML Challenge loan decision data [FICO, 2018] shown in Table 5 demonstrates that altering the reference point may increase the average SEV$^-$ (from 3 to 5), but also introduces "unexplainable" samples (meaning SEV$^-\geq$10). Hence, SEV$^-$ is sensitive to the reference.

Table 5: SEV$^-$ change by moving reference point $\tilde{r}$ moving closer to the decision boundary to $\tilde{r}'$

| MODEL | REFERENCE POINT | MEAN SEV$^-$ | % OF SAMPLES | | |
|---|---|---|---|---|---|
| | | | SEV $\geq 3$ | SEV $\geq 6$ | SEV $\geq 10$ |
| L2LR | $\tilde{r}$ | 2.76 | 2.82 | 0 | 0 |
| | $\tilde{r}'$ | 4.95 | 89.23 | 32.3 | 0 |
| L1LR | $\tilde{r}$ | 2.46 | 1.00 | 0 | 0 |
| | $\tilde{r}'$ | 4.57 | 56.87 | 21.27 | 0 |

## C Detailed Description for Score-based Soft K-Means

As we have discussed in Section 4.1, $\text{SEV}^-$ needs to have negatively predicted reference points. Therefore, when clustering the negative population, it is necessary to avoid positively predicted cluster centers. However, for most of the existing clustering methods, it is hard to "penalize" the positive predicted clusters, or their assigned samples. Therefore, we have modified the soft K-Means [Bezdek et al., 1984] algorithm so as to encourage negative clustering results.

The original Soft K-Means (SKM) algorithm generalizes K-means clustering by assigning membership scores for multiple clusters to each point. Given a data set $X = \{\boldsymbol{x}_1, \boldsymbol{x}_2, \cdots, \boldsymbol{x}_n\}$ and $C$ clusters, the goal is to minimize the objective function $J(U, V)$, where $U = [u_{ij}]$ is the membership matrix and $V = \{\mathbf{v}_1, \cdots, \mathbf{v}_C\}$ are the weighted cluster centroids. The objective is to minimize:

$$J(U, V) = \sum_{i=1}^{n} \sum_{j=1}^{C} u_{ij}^m \|\boldsymbol{x}_i - \mathbf{v}_j\|_2^2 \tag{5}$$

where $u_{ij}$ is the (soft) membership score of $\boldsymbol{x}_i$ in cluster $j$:

$$u_{i,j} = \frac{1}{\sum_{k=1}^{C} \left( \frac{\|x_i - v_j\|_2}{\|x_i - c_k\|_2} \right)^{\frac{2}{m-1}}} \tag{6}$$

and $m > 1$ is a parameter that controls the strength towards each neighboring point. When $m \approx 1$, the SKM is similar to the performance of hard K-means clustering methods. When $m > 1$ for point $i$, it is considered to be associated with multiple clusters instead of one distinct cluster. The higher the value of $m$, the more a point is considered to be part of multiple clusters, thereby reducing the distinctness of each cluster and creating a more integrated and interconnected clustering arrangement. To avoid the cluster group being predicted positively, we have given higher $m$ for those positive samples. Therefore, if the samples are predicted as positive, it reduces the possibility that those positively predicted samples to group as a cluster, which we can replace $m$ as $m_i'$ for each instance $\boldsymbol{x}_i$ as

$$m_i' = 2m \cdot \min\{f(\boldsymbol{x}_i) - 0.5, 0\} + 1. \tag{7}$$

The value of $\min\{f(\boldsymbol{x}_i) - 0.5, 0\}$ increases as $\boldsymbol{x}_i$ is classified as positive and further away from the decision boundary. As $m'$ increases, the negatively predicted samples are more associated with one distinct cluster, while the positively predicted samples are associated with multiple clusters with smaller weight. This makes the cluster centers less likely to be influenced by positively predicted points. Thus, we can rewrite the objective of the soft K-Means algorithm can be modified as

$$J'(U, V) = \sum_{i=1}^{n} \sum_{j=1}^{C} u_{ij}^{m_i'} \|\boldsymbol{x}_i - \mathbf{v}_j\|_2^2. \tag{8}$$

We call this new objective function for encouraging negative clustering centers Score-based Soft K-Means (SSKM). In our experiments, the clustering is applied to the dataset after PaCMAP [Wang et al., 2021], and the feature mean of all samples in a cluster is considered as the cluster center of this cluster, which is eventually used as a reference point. The queries are assigned to reference points that are closest (based on $\ell_2$ distance) to them in the PaCMAP embedding space for $\text{SEV}^{\copyright}$ calculation. The reason why we would like to first embed the dataset is that the dimension of the datasets might be too high for direct clustering, and PaCMAP provides an embedding that preserves both local and global structure. Figure 9 shows the probability of the negative predicted instances, as well as the clustering results using different kinds of clustering methods. The red points and stars represent the positively predicted instances and cluster centers, while the blue ones are the negatively predicted instances and cluster centers. It is evident from the Figure that that SKM is more likely to introduce positively predicted cluster centers, compared to SSKM.

When we calculate $\text{SEV}^{\copyright}$ in the experiments, all clustering parameters are tuned and fixed. For the rest of the datasets, the embedding using PaCMAP, and their clustering results for the negative population with their cluster centers, are shown in Figure 10. The regions with different colors represent different clusters, the blue stars in the graphs are cluster centers, and the gray points within the graphs are positive queries. All those cluster centers can be constrained to be predicted as negative by tuning the hyperparameter for Score-based Soft K-Means. Note that if one of the cluster centers cannot be constrained to be predicted as negative even with high $m$, then it is reasonable to remove this cluster center when calculating $\text{SEV}^{\copyright}$.

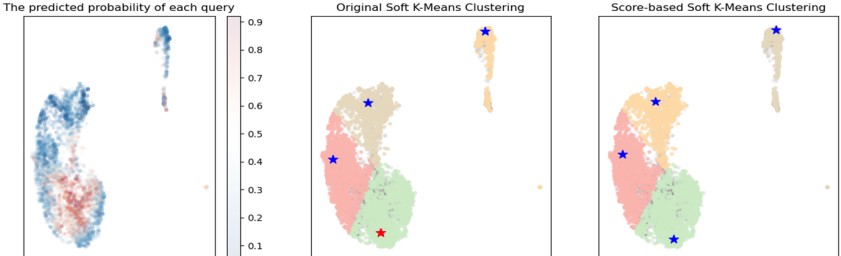

Figure 9: The clustering results for FICO dataset. (Left) The probability distribution for the negatively labeled queries; (Middle) The clustering result for Original Soft K-Means Clustering; (Right) The clustering result for Score-based K-Means Clustering The red stars represent the positively predicted cluster centers, and the blue stars the negatively predicted cluster centers

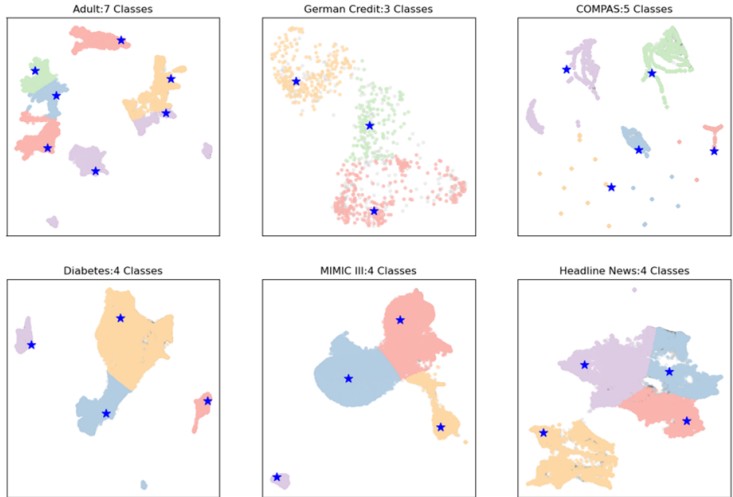

Figure 10: Clustering Results for different datasets.

# D Detailed Algorithm for Flexible-based SEV

This section presents how the flexible-based SEV ($\text{SEV}^F$) has done to determine the flexible references. The key idea of finding the reference is to do a grid search through each of the features in the training dataset based on the original reference, and find the feature values that has the minimum model outcome.

---

**Algorithm 1** Reference Search for Flexible SEV

---

1: **Input:** The negative samples $X^-$, flexibility $\epsilon$, reference $\tilde{\boldsymbol{r}}$, grid size $G$
2: **Output:** Flexible reference $\tilde{\boldsymbol{r}}'$
3: **Initialization**: $\tilde{\boldsymbol{r}}' \leftarrow \tilde{\boldsymbol{r}}$
4: **for** each feature $j \in \mathcal{J}$, where $\tilde{r}_j$ is the reference value of feature $j$ in $X^-$ **do**
5:     $q_j \leftarrow \text{quantile}(X_j^-, \tilde{r}_j)$    {Quantile location of $\tilde{r}_j$}
6:     $B_j^+ \leftarrow \text{percentile}(X_j^-, q_j + \epsilon)$   {The upper range}
7:     $B_j^- \leftarrow \text{percentile}(X_j^-, q_j - \epsilon)$    {The lower range}
8:     $B_j^{(g)} \sim \text{Uniform}[B_j^-, B_j^+], g = 1, \cdots, G$
9:     $P_j^{(g)} \leftarrow f([\tilde{r}_1, \cdots, B_j^{(g)}, \cdots \tilde{r}_J]), g = 1 \cdots G$ {Slight change to feature $j$ for prediction}
10:    $g' \leftarrow \arg\min_g P_j^{(g)}$ {Find minimum model outcome}
11:    $\tilde{r}'_j \leftarrow B_j^{(g')}$ {Update for flexible references}
12: **end for**

---

# E  Detailed Algorithms for Tree-based SEV

This section presents how the tree-based SEV is calculated through two main procedure: Algorithm 2 (Preprocessing) for collecting all negative pathways and assigning them to each internal nodes and Algorithm 3 (Efficient $\text{SEV}^T$ Calculation) for checking all negative pathways conditions for each query and calculating the number of feature changes.

---

**Algorithm 2** Preprocessing - Information collection process for $\text{SEV}^T$

---

1: **Input:** Decision tree $DT$
2: **Output:** $DT^-$, a dictionary of paths to negative predictions for each internal node encoding
3: $nodes \leftarrow [DT.root]$
4: $negative\_path \leftarrow []$
5: {Negative path collection procedure}
6: **while** $nodes$ not empty **do**
7:    $[node, path] \leftarrow nodes.pop()$
8:    **if** $node$ is a negative leaf **then**
9:      $negative\_path.append(path)$
10:    **else if** $node$ is an internal node or a root node **then**
11:      {A}dd the child nodes and the path to the node list
12:      $nodes.append([node.left, path+\text{“}L\text{”}])$
13:      $nodes.append([node.right, path+\text{“}R\text{”}])$
14:    **else**
15:      Continue {if the leaf is positive, ignore it}
16:    **end if**
17: **end while**
18: {Assign Negative Pathways to root or internal nodes}
19: $DT^- \leftarrow \text{dict}()$
20: **for** each $path \; in \; negative\_path$ **do**
21:    **for** $i = 1, \cdots path.length$ **do**
22:      {Add the negative decision path for internal nodes}
23:      $curr\_node \leftarrow negative\_path[:i]$
24:      {$curr\_node$ is the encoded internal node, and $negative \; path[i:]$ is a negative decision path below this node}
25:      $DT^-[curr\_node].append(negative\_path[i:])$
26:    **end for**
27: **end for**

---

**Algorithm 3** Efficient SEV$^T$ Calculation – Negative Pathways Check

---

1: **Input:** $DT$: decision tree, $DT^-$: decision trees with paths to negative predictions, query value $x_i$, $DP_i$: list of internal nodes representing decision process for $x_i$, $path_i$: the encoded $DP_i$
2: **Output:** SEV$^T$
3: **INITIALIZATION:** SEV$^T \leftarrow 0$
4: *decision_path* $\leftarrow$ encoded($DT$, $x_i$)
5: {encoded($DT$, $x_i$) is a function to get the string representation of the query $x_i$ or a node *node* for $DT$, e.g. "LR","LL" mentioned in section 4.2}
6: **for** each internal node *node* in $DP_i$ **do**
7:     **if** *node* has a sibling leaf node and is predicted as negative **then**
8:         SEV$^T \leftarrow 1$ {Based on Theoerem 4.1}
9:         Break {SEV$^T$=1 is the smallest SEV$^T$, no further calculation needed}
10:     **end if**
11:     *encoded_node* $\leftarrow$ encoded($DT$, *node*) {Get the string representation of *node*}
12:     *negative_paths* $\leftarrow DT^-$[encoded_node] {Get the negative pathways *encoded_node* have}
13:     **for** each *path* in *negative_path* **do**
14:         {If the negative goes the same direction as the decision path, we don't need to calculate this path again}
15:         {*path*[0] is the first character in *path*}
16:         **if** *decision_path*[encoded_node.length]=*path*[0] **then**
17:             Continue
18:         **end if**
19:         *temp_sev* $\leftarrow 0$
20:         {Go over the condition in the *path*}
21:         {Check if query $x_i$ satisfies, if it doesn't satisfies the condition, then *temp_sev* should add 1}

22:         **for** *condition* in each *path* **do**
23:             **if** $x_i$ doesn't satisfy *condition* AND $x_i$ hasn't been changed yet **then**
24:                 *temp_sev* $\leftarrow$ *temp_sev* $+1$
25:             **end if**
26:         **end for**
27:         SEV$^T \leftarrow \min\{$*temp_sev*, SEV$^T\}$ {Update SEV$^T$ to be the samller one}
28:         **if** SEV$^T = 1$ **then**
29:             Break {SEV$^T$=1 is the smallest SEV$^T$, no further calculation needed}
30:         **end if**
31:     **end for**
32: **end for**

---

# F Model training and parameter selection

Baseline models were fit using `sklearn` [Pedregosa et al., 2011] implementations in Python. The logistic regression models L1 LR and L2 LR were fit using regularization parameter $C = 0.01$. The 2-layer MLP used ReLU activation and consisted of two fully-connected layers with 128 nodes each. It was trained with early stopping. The gradient-boosted classifier used 200 trees with a max depth of 3. For tree-based methods comparisons, the decision tree classifiers were fit using `sklearn` [Pedregosa et al., 2011] and TreeFARMS packages [Wang et al., 2022b]. Since GOSDT methods require binary input, we used the built-in threshold guessing function in GOSDT to binarize the features with set of parameters `n_est=50`, and `max_depth=1`. All the models are trained using a RTX2080Ti GPU, and with 4 core in Intel(R) Xeon(R) Gold 6226 CPU @ 2.70GHz.

In order to test the performance of All-Opt$^-$, all models mentioned above were trained by adding the SEV losses from Section 5 to the standard loss term (`BCELoss`). For GBDT, the training goal is to reweigh the trees from the baseline GBDT model. The resulting loss was minimized via gradient descent in PyTorch [Paszke et al., 2019], with a batch size of 128, a learning rate of 0.1, and the Adam optimizer. To maintain high accuracy, the first 80 training epochs are warm-up epochs optimizing just Binary Cross Entropy Loss for classification (`BCELoss`). The next 20 epochs add the All-Opt terms and the baseline positive penalty term to encourage low SEV values. Moreover, during the optimization process, it is important to ensure that the reference has a negative prediction. If the reference is predicted as positive, then the SEV$^-$ may not exist, and a sparse explanation is no longer meaningful. Thus, we add a term to penalize the reference if it receives a positive prediction:

$$\ell_{\text{Pos\_ref}}(f) := \sum_{i=1}^{n} \max(f(\tilde{\boldsymbol{r}}_i), 0.5 - \theta)$$

where $\theta > 0$ is a margin parameter, usually $\theta = 0.05$. This term is $(0.5 - \theta)$ as long as the reference is predicted negative. As soon as it exceeds that amount, it is penalized (increasing linearly in $f(\tilde{\boldsymbol{r}})$).

To put these into an algorithm, we optimize a linear combination of different loss terms,

$$\min_{f \in \mathcal{F}} \ell_{\text{BCE}}(f) + C_1 \ell_{\text{SEV\_All\_Opt}-}(f) + C_2 \ell_{\text{Pos\_ref}}(f) \tag{9}$$

Therefore, we are tuning both $C_1$ and $C_2$ to find a model with sparser explanations without performance loss through grid search. For cluster-based SEV, the cluster centers are recalculated based on the new model every 5 epochs.

# G The sparsity and meaningful performance of different counterfactual explanation methods

In this section, we provide detailed information on other kinds of counterfactual explanations generated by the `CARLA` package [Pawelczyk et al., 2021] on different datasets for logistic regression models. Table 6 shows the number of features changed and the $\ell_\infty$ for different counterfactual explanations. These counterfactual explanations tend to provide less sparse explanations than other SEV$^-$ variants shown in Section 6.3. For the $\ell_\infty$ calculations, we consider only the numerical features, since the categorical features' $\ell_\infty$ norm does not provide meaningful explanations. Moreover, we have calculated the average log-likelihood of the explanations using the Gaussian Mixture Model in scikit-learn Pedregosa et al. [2011]. The parameter `n_components` for each dataset is selected based on the clustering result mentioned in Appendix C. Here, we are using the same Gaussian Mixture Model for evaluating whether the explanation is within a high-density region.

Table 6: Explanation performance in different counterfactual explanations

| DATASET | COUNTERFACTUAL EXPLANATIONS | MEAN $\ell_\infty$ | # FEATURES CHANGE | MEDIAN LOG-LIKELIHOOD |
|---|---|---|---|---|
| Adult | Growing Sphere | $1.07 \pm 0.01$ | $14 \pm 0.00$ | $345.03 \pm 34.19$ |
| | DiCE | $0.78 \pm 0.02$ | $2.19 \pm 0.12$ | $-24752.12 \pm 452.47$ |
| | REVISE | $6.1 \pm 0.02$ | $12.14 \pm 0.75$ | $345.03 \pm 32.84$ |
| | Watcher | $0.01 \pm 0.01$ | $6.00 \pm 0.00$ | $345.12 \pm 34.19$ |
| | SEV[1] | $22.62 \pm 0.01$ | $1.18 \pm 0.02$ | $-24752.12 \pm 452.47$ |
| | SEV[©] | $2.86 \pm 0.01$ | $1.34 \pm 0.02$ | $156.88 \pm 59.67$ |
| COMPAS | Growing Sphere | $0.02 \pm 0.01$ | $7.00 \pm 0.00$ | $10.47 \pm 0.00$ |
| | DiCE | $1.38 \pm 0.02$ | $3.20 \pm 0.45$ | $-6.68 \pm 0.02$ |
| | REVISE | $1.12 \pm 0.03$ | $5.54 \pm 0.63$ | $-1.84 \pm 0.21$ |
| | Watcher | $0.01 \pm 0.01$ | $5.00 \pm 0.00$ | $10.48 \pm 0.03$ |
| | SEV[1] | $2.31 \pm 0.01$ | $1.22 \pm 0.02$ | $14.65 \pm 0.32$ |
| | SEV[©] | $2.06 \pm 0.01$ | $1.19 \pm 0.02$ | $14.41 \pm 0.05$ |
| Diabetes | Growing Sphere | $0.01 \pm 0.01$ | $33.00 \pm 0.00$ | $320.41 \pm 21.47$ |
| | DiCE | $0.71 \pm 0.12$ | $2.76 \pm 0.15$ | $-74296.98 \pm 861.27$ |
| | REVISE | $0.80 \pm 0.02$ | $15.84 \pm 0.02$ | $320.41 \pm 16.73$ |
| | Watcher | $0.01 \pm 0.01$ | $12 \pm 0.00$ | $320.41 \pm 21.34$ |
| | SEV[1] | $2.7 \pm 0.10$ | $1.63 \pm 0.01$ | $309.56 \pm 15.32$ |
| | SEV[©] | $2.31 \pm 0.12$ | $1.28 \pm 0.02$ | $320.71 \pm 14.79$ |
| FICO | Growing Sphere | $0.01 \pm 0.01$ | $23 \pm 0.00$ | $-10.93 \pm 0.42$ |
| | DiCE | $1.15 \pm 0.13$ | $3.27 \pm 0.17$ | $-20.11 \pm 0.3$ |
| | REVISE | $0.12 \pm 0.01$ | $23 \pm 0.00$ | $-10.94 \pm 0.42$ |
| | Watcher | $0.01 \pm 0.01$ | $23 \pm 0.00$ | $-10.94 \pm 0.41$ |
| | SEV[1] | $1.81 \pm 0.01$ | $2.76 \pm 0.02$ | $-20.11 \pm 0.32$ |
| | SEV[©] | $1.82 \pm 0.01$ | $2.21 \pm 0.02$ | $-19.32 \pm 0.21$ |
| German Credit | Growing Sphere | $0.01 \pm 0.02$ | $20 \pm 0.00$ | $52.20 \pm 0.02$ |
| | DiCE | $6.08 \pm 0.01$ | $2.76 \pm 0.23$ | $-53908.78 \pm 367.84$ |
| | REVISE | $0.16 \pm 0.01$ | $7.65 \pm 0.12$ | $-73492.06 \pm 492.45$ |
| | Watcher | $0.01 \pm 0.00$ | $6.00 \pm 0.00$ | $52.23 \pm 0.04$ |
| | SEV[1] | $3.08 \pm 0.01$ | $1.51 \pm 0.02$ | $-124914.32 \pm 792.52$ |
| | SEV[©] | $3.2 \pm 0.01$ | $1.17 \pm 0.02$ | $50.21 \pm 0.32$ |
| Headline | Growing Sphere | $0.01 \pm 0.00$ | $18 \pm 0.00$ | $-4.56 \pm 0.02$ |
| | DiCE | $1.13 \pm 0.02$ | $2.79 \pm 0.14$ | $-12.84 \pm 0.42$ |
| | REVISE | $1.81 \pm 0.13$ | $15.93 \pm 0.24$ | $-6.98 \pm 0.12$ |
| | Watcher | $0.01 \pm 0.01$ | $12 \pm 0.00$ | $-4.56 \pm 0.02$ |
| | SEV[1] | $2.50 \pm 0.02$ | $1.98 \pm 0.01$ | $1.52 \pm 0.12$ |
| | SEV[©] | $2.94 \pm 0.02$ | $1.62 \pm 0.02$ | $0.89 \pm 0.26$ |
| MIMIC | Growing Sphere | $0.01 \pm 0.01$ | $14 \pm 0.00$ | $-24.52 \pm 0.02$ |
| | DiCE | $1.34 \pm 0.23$ | $6.47 \pm 0.24$ | $-26.55 \pm 0.02$ |
| | REVISE | $0.01 \pm 0.00$ | $12 \pm 0.00$ | $-24.52 \pm 0.01$ |
| | Watcher | $0.01 \pm 0.00$ | $12 \pm 0.00$ | $-24.52 \pm 0.01$ |
| | SEV[1] | $4.53 \pm 0.49$ | $1.18 \pm 0.02$ | $-20.11 \pm 0.32$ |
| | SEV[©] | $1.98 \pm 0.13$ | $1.19 \pm 0.02$ | $-19.32 \pm 0.15$ |

# H Detailed SEV$^-$ for all datasets

In this section, we show how SEV$^1$, SEV$^{\circledcirc}$, SEV$^{\circledcirc+F}$ can increase the similarity metrics or reduce the sparsity explanations. All the models are trained and evaluated 10 times using different splits, and evaluated for their mean SEV$^-$, mean $\ell_\infty$, as well as their explanation time for each query.

Table 7 shows the model performance and SEV$^1$ on various datasets. SEV$^1$ is considered as a base case for other SEV$^-$ variants to compare with. Table 7 shows that SEV$^1$ yields very high $\ell_\infty$ for each model, indicating a large distance between the query and reference, which implies low closeness according to Section 3.2.

Table 8 shows the model performance and SEV$^{\circledcirc}$ on different datasets. Similarly, The Mean SEV$^{\circledcirc}$ column reports the mean SEV$^{\circledcirc}$ for the model and the decrease in mean SEV$^-$ in percentage compared to SEV$^1$ (reported in the parenthesis). The Mean $\ell_\infty$ column reports the mean $\ell_\infty$ and the percentage reduction compared to SEV$^1$. On most datasets, SEV$^{\circledcirc}$ increases, and $\ell_\infty$ decreases, which means that the model is providing both sparser and more meaningful explanations. For some datasets like Adult and MIMIC, the SEV$^{\circledcirc}$ increases, since the cluster-based reference points might be closer to the decision boundary of the model as each query is trying to find the closest (in $\ell_2$ distance) negatively predicted reference point, which might provide less sparse explanations.

Table 9 shows the model performance and SEV$^{\circledcirc+F}$ (SEV$^{\circledcirc}$ with variable reference) on various datasets with different flexibility levels. The Mean SEV$^F$ column reports the mean SEV$^-$ for the model and the decrease in mean SEV$^-$ in percentage compared to SEV$^1$ (reported in the parenthesis). The Mean $\ell_\infty$ column reports the mean $\ell_\infty$ and the percentage reduction compared to SEV$^1$. It is evident that with SEV$^F$, SEV$^-$ decreases, but the $\ell_\infty$ norm will increase due to the flexibility of the features mentioned in section 4.4. The "flexibility used" column shows the proportion of queries using the flexible reference instead of the original one for calculating SEV$^F$, and the higher the proportion, the larger decrease in SEV$^-$ the model can achieve.

Table 7: The SEV$^1$ under different models

| Dataset | Model | Train Accuracy | Test Accuracy | Train AUC | Test AUC | Average SEV$^1$ | Median $\ell_\infty$ | Explanation Time($10^{-2}$s) | Average Log-likelihood |
|---|---|---|---|---|---|---|---|---|---|
| Adult | GBDT | 0.88 ± 0.0 | 0.87 ± 0.0 | 0.93 ± 0.0 | 0.93 ± 0.0 | 1.23 ± 0.02 | 18.28 ± 1.8 | 0.69 ± 0.08 | −57437.86 ± 2718.7 |
| | L1LR | 0.85 ± 0.0 | 0.85 ± 0.0 | 0.9 ± 0.0 | 0.9 ± 0.0 | 1.14 ± 0.01 | 24.2 ± 2.41 | 0.26 ± 0.01 | −44735.07 ± 1393.91 |
| | L2LR | 0.85 ± 0.0 | 0.85 ± 0.0 | 0.9 ± 0.0 | 0.9 ± 0.0 | 1.18 ± 0.0 | 22.62 ± 2.27 | 0.16 ± 0.01 | −49293.12 ± 1157.19 |
| | MLP | 0.87 ± 0.0 | 0.86 ± 0.0 | 0.93 ± 0.0 | 0.92 ± 0.0 | 1.27 ± 0.06 | 21.73 ± 3.57 | 0.62 ± 0.17 | −67000.48 ± 5030.26 |
| COMPAS | GBDT | 0.7 ± 0.0 | 0.67 ± 0.01 | 0.77 ± 0.0 | 0.72 ± 0.01 | 1.15 ± 0.04 | 1.94 ± 0.08 | 0.18 ± 0.24 | 8.15 ± 0.97 |
| | L1LR | 0.68 ± 0.0 | 0.67 ± 0.01 | 0.73 ± 0.0 | 0.72 ± 0.01 | 1.25 ± 0.02 | 2.31 ± 0.07 | 0.12 ± 0.0 | 5.09 ± 0.92 |
| | L2LR | 0.68 ± 0.0 | 0.67 ± 0.02 | 0.73 ± 0.0 | 0.72 ± 0.01 | 1.26 ± 0.03 | 2.41 ± 0.09 | 0.08 ± 0.01 | 5.19 ± 1.0 |
| | MLP | 0.69 ± 0.01 | 0.67 ± 0.01 | 0.74 ± 0.01 | 0.72 ± 0.01 | 1.35 ± 0.12 | 2.3 ± 0.32 | 0.27 ± 0.09 | 6.49 ± 1.1 |
| Diabetes | GBDT | 0.65 ± 0.0 | 0.64 ± 0.0 | 0.66 ± 0.0 | 0.66 ± 0.0 | 1.39 ± 0.01 | 2.82 ± 0.01 | 364.74 ± 92.38 | −59814.81 ± 2356.74 |
| | L1LR | 0.62 ± 0.0 | 0.62 ± 0.0 | 0.66 ± 0.0 | 0.66 ± 0.0 | 1.62 ± 0.01 | 2.6 ± 0.01 | 106.63 ± 79.76 | −20834.12 ± 1378.32 |
| | L2LR | 0.62 ± 0.0 | 0.62 ± 0.0 | 0.66 ± 0.0 | 0.66 ± 0.0 | 1.63 ± 0.01 | 2.7 ± 0.01 | 117.63 ± 79.76 | −19117.45 ± 1091.56 |
| | MLP | 0.65 ± 0.01 | 0.64 ± 0.0 | 0.71 ± 0.01 | 0.69 ± 0.0 | 1.69 ± 0.13 | 2.67 ± 0.09 | 136.33 ± 140.47 | −70595.3 ± 3666.52 |
| FICO | GBDT | 0.71 ± 0.0 | 0.7 ± 0.0 | 0.78 ± 0.0 | 0.77 ± 0.0 | 3.58 ± 0.12 | 1.81 ± 0.01 | 692.83 ± 30.77 | −74.13 ± 8.92 |
| | L1LR | 0.71 ± 0.0 | 0.7 ± 0.0 | 0.78 ± 0.0 | 0.77 ± 0.01 | 2.47 ± 0.11 | 1.81 ± 0.07 | 100.83 ± 30.77 | −81.31 ± 7.41 |
| | L2LR | 0.72 ± 0.0 | 0.71 ± 0.01 | 0.78 ± 0.0 | 0.78 ± 0.01 | 2.76 ± 0.12 | 1.93 ± 0.04 | 481.75 ± 146.53 | −52.09 ± 2.1 |
| | MLP | 0.72 ± 0.01 | 0.71 ± 0.01 | 0.8 ± 0.02 | 0.78 ± 0.01 | 2.7 ± 0.29 | 1.88 ± 0.15 | 553.15 ± 463.34 | −67.71 ± 13.05 |
| German Credit | GBDT | 0.96 ± 0.01 | 0.75 ± 0.02 | 0.99 ± 0.0 | 0.77 ± 0.02 | 1.39 ± 0.12 | 1.87 ± 0.46 | 2.69 ± 1.8 | −75811.5 ± 6476.74 |
| | L1LR | 0.75 ± 0.01 | 0.75 ± 0.01 | 0.8 ± 0.01 | 0.79 ± 0.05 | 1.3 ± 0.06 | 2.45 ± 0.16 | 0.78 ± 0.49 | −64237.32 ± 26906.43 |
| | L2LR | 0.78 ± 0.01 | 0.76 ± 0.03 | 0.83 ± 0.01 | 0.79 ± 0.04 | 1.51 ± 0.15 | 3.08 ± 0.42 | 1.34 ± 0.96 | −111945.26 ± 9916.8 |
| | MLP | 0.81 ± 0.04 | 0.76 ± 0.03 | 0.87 ± 0.04 | 0.78 ± 0.04 | 1.6 ± 0.19 | 2.69 ± 0.45 | 7.68 ± 5.59 | −119557.08 ± 15328.57 |
| Headline | GBDT | 0.82 ± 0.0 | 0.81 ± 0.0 | 0.9 ± 0.0 | 0.89 ± 0.0 | 1.82 ± 0.03 | 2.35 ± 0.02 | 16.25 ± 2.45 | −395.41 ± 340.77 |
| | L1LR | 0.78 ± 0.0 | 0.78 ± 0.0 | 0.85 ± 0.0 | 0.85 ± 0.0 | 1.92 ± 0.01 | 2.51 ± 0.02 | 6.73 ± 0.38 | −558.81 ± 287.68 |
| | L2LR | 0.78 ± 0.0 | 0.78 ± 0.0 | 0.86 ± 0.0 | 0.85 ± 0.0 | 1.98 ± 0.01 | 2.5 ± 0.02 | 9.21 ± 0.49 | −555.95 ± 286.15 |
| | MLP | 0.83 ± 0.01 | 0.81 ± 0.0 | 0.91 ± 0.01 | 0.89 ± 0.0 | 2.03 ± 0.03 | 2.31 ± 0.07 | 26.25 ± 2.45 | −493.37 ± 316.22 |
| MIMIC | GBDT | 0.91 ± 0.0 | 0.9 ± 0.0 | 0.87 ± 0.0 | 0.85 ± 0.0 | 1.18 ± 0.02 | 1.28 ± 0.15 | 1.03 ± 0.22 | −18.92 ± 0.37 |
| | L1LR | 0.89 ± 0.0 | 0.89 ± 0.0 | 0.8 ± 0.0 | 0.8 ± 0.0 | 1.15 ± 0.04 | 4.53 ± 0.49 | 0.26 ± 0.04 | −19.76 ± 0.52 |
| | L2LR | 0.89 ± 0.0 | 0.89 ± 0.0 | 0.8 ± 0.0 | 0.8 ± 0.0 | 1.16 ± 0.02 | 4.34 ± 0.52 | 0.29 ± 0.03 | −19.66 ± 0.49 |
| | MLP | 0.9 ± 0.0 | 0.9 ± 0.0 | 0.87 ± 0.01 | 0.85 ± 0.0 | 1.18 ± 0.03 | 2.08 ± 0.35 | 0.79 ± 0.19 | −17.25 ± 0.84 |

Table 8: The SEV$^{©}$ under different models

| DATASET | MODEL | TRAIN ACCURACY | TEST ACCURACY | TRAIN AUC | TEST AUC | AVERAGE SEV | MEDIAN $\ell_\infty$ | AVERAGE TIME $(10^{-2})$ | AVERAGE LOG-LIKELIHOOD |
|---------|-------|------|------|------|------|------|------|------|------|
| Adult | GBDT | $0.88 \pm 0.0$ | $0.87 \pm 0.0$ | $0.93 \pm 0.0$ | $0.93 \pm 0.0$ | 1.39(13.01%) | 2.41(-86.82%) | $2.22 \pm 0.84$ | $-22974.51(60.0\%)$ |
| | L1LR | $0.85 \pm 0.0$ | $0.85 \pm 0.0$ | $0.9 \pm 0.0$ | $0.9 \pm 0.0$ | 1.23(7.89%) | 2.05(-91.53%) | $0.56 \pm 0.03$ | $-39333.37(12.07\%)$ |
| | L2LR | $0.85 \pm 0.0$ | $0.85 \pm 0.0$ | $0.9 \pm 0.0$ | $0.9 \pm 0.0$ | 1.34(13.56%) | 2.86(-87.36%) | $0.38 \pm 0.12$ | $-21033.54(57.33\%)$ |
| | MLP | $0.87 \pm 0.0$ | $0.86 \pm 0.0$ | $0.93 \pm 0.0$ | $0.92 \pm 0.0$ | 1.62(27.56%) | 5.16(-76.25%) | $1.18 \pm 0.53$ | $-23421.5(60.97\%)$ |
| COMPAS | GBDT | $0.7 \pm 0.0$ | $0.67 \pm 0.01$ | $0.77 \pm 0.0$ | $0.72 \pm 0.01$ | 1.18(2.61%) | 1.52(-21.65%) | $0.32 \pm 0.03$ | 9.08(11.41%) |
| | L1LR | $0.68 \pm 0.0$ | $0.67 \pm 0.01$ | $0.73 \pm 0.0$ | $0.72 \pm 0.01$ | 1.19(-4.8%) | 1.75(-24.24%) | $0.12 \pm 0.01$ | 5.53(8.64%) |
| | L2LR | $0.68 \pm 0.0$ | $0.67 \pm 0.02$ | $0.73 \pm 0.0$ | $0.72 \pm 0.01$ | 1.22(-3.17%) | 2.06(-14.52%) | $0.09 \pm 0.01$ | 5.98(15.22%) |
| | MLP | $0.69 \pm 0.01$ | $0.67 \pm 0.01$ | $0.74 \pm 0.01$ | $0.72 \pm 0.01$ | 1.3(-3.7%) | 1.82(-20.87%) | $0.15 \pm 0.03$ | 9.12(40.52%) |
| Diabetes | GBDT | $0.65 \pm 0.0$ | $0.64 \pm 0.0$ | $0.7 \pm 0.0$ | $0.7 \pm 0.0$ | 1.36(-2.21%) | 1.89(-49.21%) | $17.39 \pm 7.21$ | $-5572.49(90.55\%)$ |
| | L1LR | $0.62 \pm 0.0$ | $0.62 \pm 0.0$ | $0.66 \pm 0.0$ | $0.66 \pm 0.0$ | 1.22(-24.6%) | 2.31(-11.58%) | $2.1 \pm 0.4$ | $-5460.38(92.27\%)$ |
| | L2LR | $0.62 \pm 0.0$ | $0.62 \pm 0.0$ | $0.66 \pm 0.0$ | $0.66 \pm 0.0$ | 1.28(-21.47%) | 2.31(-14.44%) | $3.8 \pm 1.26$ | $-14461.36(24.36\%)$ |
| | MLP | $0.65 \pm 0.0$ | $0.63 \pm 0.0$ | $0.7 \pm 0.01$ | $0.69 \pm 0.0$ | 1.47(-13.02%) | 2.24(-16.1%) | $23.28 \pm 14.31$ | $-11320.72(83.96\%)$ |
| FICO | GBDT | $0.77 \pm 0.0$ | $0.72 \pm 0.01$ | $0.85 \pm 0.0$ | $0.79 \pm 0.01$ | 2.06(-42.52%) | 1.08(-40.3%) | $23.34 \pm 8.86$ | $-59.52(19.7\%)$ |
| | L1LR | $0.71 \pm 0.0$ | $0.7 \pm 0.0$ | $0.78 \pm 0.0$ | $0.77 \pm 0.0$ | 1.79(-27.53%) | 1.95(7.73%) | $3.11 \pm 1.02$ | $-77.53(4.65\%)$ |
| | L2LR | $0.72 \pm 0.0$ | $0.71 \pm 0.01$ | $0.78 \pm 0.0$ | $0.77 \pm 0.01$ | 2.21(-19.93%) | 1.82(-5.7%) | $39.49 \pm 16.49$ | $-58.86(-13.0\%)$ |
| | MLP | $0.74 \pm 0.01$ | $0.71 \pm 0.01$ | $0.81 \pm 0.01$ | $0.78 \pm 0.01$ | 2.15(-20.37%) | 1.75(-6.91%) | $26.26 \pm 9.01$ | $-62.6(7.55\%)$ |
| German Credit | GBDT | $0.96 \pm 0.01$ | $0.75 \pm 0.02$ | $0.99 \pm 0.0$ | $0.77 \pm 0.03$ | 1.22(-12.23%) | 1.73(-7.49%) | $0.79 \pm 0.53$ | $-28478.65(62.43\%)$ |
| | L1LR | $0.75 \pm 0.01$ | $0.75 \pm 0.02$ | $0.8 \pm 0.01$ | $0.77 \pm 0.04$ | 1.03(-20.77%) | 1.52(-37.96%) | $0.05 \pm 0.01$ | $-23691.73(63.12\%)$ |
| | L2LR | $0.78 \pm 0.01$ | $0.76 \pm 0.03$ | $0.83 \pm 0.01$ | $0.79 \pm 0.04$ | 1.17(-22.52%) | 3.2(3.9%) | $0.1 \pm 0.07$ | $-40622.35(63.71\%)$ |
| | MLP | $0.81 \pm 0.04$ | $0.76 \pm 0.03$ | $0.87 \pm 0.04$ | $0.78 \pm 0.04$ | 1.24(-22.5%) | 2.54(-5.58%) | $0.24 \pm 0.2$ | $-40045.69(66.5\%)$ |
| Headline | GBDT | $0.82 \pm 0.0$ | $0.81 \pm 0.0$ | $0.9 \pm 0.0$ | $0.89 \pm 0.0$ | 1.76(-3.3%) | 2.18(-7.23%) | $6.96 \pm 0.84$ | $-383.24(-3.08\%)$ |
| | L1LR | $0.78 \pm 0.0$ | $0.78 \pm 0.0$ | $0.85 \pm 0.0$ | $0.85 \pm 0.0$ | 1.57(-18.23%) | 2.94(17.13%) | $0.88 \pm 0.21$ | $-559.35(0.1\%)$ |
| | L2LR | $0.78 \pm 0.0$ | $0.78 \pm 0.0$ | $0.86 \pm 0.0$ | $0.85 \pm 0.0$ | 1.62(-18.18%) | 2.94(17.6%) | $1.46 \pm 0.1$ | $-556.52(0.1\%)$ |
| | MLP | $0.83 \pm 0.01$ | $0.81 \pm 0.0$ | $0.91 \pm 0.01$ | $0.89 \pm 0.0$ | 1.67(-17.7%) | 1.99(-16.08%) | $3.05 \pm 0.43$ | $-495.08(0.0\%)$ |
| MIMIC | GBDT | $0.91 \pm 0.0$ | $0.9 \pm 0.0$ | $0.87 \pm 0.0$ | $0.85 \pm 0.0$ | 1.21(2.54%) | 0.49(-61.72%) | $0.61 \pm 0.12$ | $-18.15(4.07\%)$ |
| | L1LR | $0.89 \pm 0.0$ | $0.89 \pm 0.0$ | $0.8 \pm 0.0$ | $0.8 \pm 0.0$ | 1.17(1.74%) | 1.8(-60.26%) | $0.17 \pm 0.03$ | $-20.41(-3.29\%)$ |
| | L2LR | $0.89 \pm 0.0$ | $0.89 \pm 0.0$ | $0.8 \pm 0.0$ | $0.8 \pm 0.0$ | 1.19(2.59%) | 1.98(-54.38%) | $0.19 \pm 0.03$ | $-20.26(-3.05\%)$ |
| | MLP | $0.9 \pm 0.0$ | $0.9 \pm 0.0$ | $0.87 \pm 0.01$ | $0.85 \pm 0.0$ | 1.23(4.24%) | 0.6(-71.15%) | $0.33 \pm 0.07$ | $-16.77(2.78\%)$ |

Table 9: SEV$^{©+F}$ under different models

| Dataset | Model | Flex-ibility | Train Accuracy | Test Accuracy | Train AUC | Test AUC | Average SEV$^-$ | Median $\ell_\infty$ | Average Log-likelihood | Explanation Time($10^{-2}$s) |
|---|---|---|---|---|---|---|---|---|---|---|
| Adult | GBDT | 0.05 | $0.88 \pm 0.0$ | $0.87 \pm 0.0$ | $0.93 \pm 0.0$ | $0.93 \pm 0.0$ | 1.3(5.69%) | 0.95(-94.8%) | −21763.14(62.11%) | $3.98 \pm 0.45$ |
| | | 0.10 | $0.88 \pm 0.0$ | $0.87 \pm 0.0$ | $0.93 \pm 0.0$ | $0.93 \pm 0.0$ | 1.29(4.88%) | 0.95(-94.8%) | −20395.38(4.49%) | $3.82 \pm 0.32$ |
| | | 0.20 | $0.88 \pm 0.0$ | $0.87 \pm 0.0$ | $0.93 \pm 0.0$ | $0.93 \pm 0.0$ | 1.29(4.88%) | 0.96(-94.75%) | −17611.65(69.34%) | $3.63 \pm 0.29$ |
| | L1LR | 0.05 | $0.85 \pm 0.0$ | $0.85 \pm 0.0$ | $0.9 \pm 0.0$ | $0.9 \pm 0.0$ | 1.2(5.26%) | 0.96(-96.03%) | −29801.44(33.38%) | $1.0 \pm 0.04$ |
| | | 0.10 | $0.85 \pm 0.0$ | $0.85 \pm 0.0$ | $0.9 \pm 0.0$ | $0.9 \pm 0.0$ | 1.19(4.39%) | 0.96(-96.03%) | −29144.93(34.85%) | $0.94 \pm 0.04$ |
| | | 0.20 | $0.85 \pm 0.0$ | $0.85 \pm 0.0$ | $0.9 \pm 0.0$ | $0.9 \pm 0.0$ | 1.19(4.39%) | 0.97(-95.99%) | −30245.09(32.39%) | $0.91 \pm 0.04$ |
| | L2LR | 0.05 | $0.85 \pm 0.0$ | $0.85 \pm 0.0$ | $0.9 \pm 0.0$ | $0.9 \pm 0.0$ | 1.32(11.86%) | 2.47(-89.08%) | −20693.31(58.02%) | $1.59 \pm 0.19$ |
| | | 0.10 | $0.85 \pm 0.0$ | $0.85 \pm 0.0$ | $0.9 \pm 0.0$ | $0.9 \pm 0.0$ | 1.32(11.86%) | 2.41(-89.35%) | −20294.61(58.83%) | $1.64 \pm 0.18$ |
| | | 0.20 | $0.85 \pm 0.0$ | $0.85 \pm 0.0$ | $0.9 \pm 0.0$ | $0.9 \pm 0.0$ | 1.32(11.86%) | 2.49(-88.99%) | −21987.43(55.39%) | $1.59 \pm 0.16$ |
| | MLP | 0.05 | $0.87 \pm 0.0$ | $0.86 \pm 0.0$ | $0.93 \pm 0.0$ | $0.92 \pm 0.0$ | 1.54(21.26%) | 2.95(-86.42%) | −27141.97(59.49%) | $3.78 \pm 1.4$ |
| | | 0.10 | $0.87 \pm 0.0$ | $0.86 \pm 0.0$ | $0.93 \pm 0.0$ | $0.92 \pm 0.0$ | 1.52(19.69%) | 2.75(-87.34%) | −23444.97(65.01%) | $3.76 \pm 1.36$ |
| | | 0.20 | $0.87 \pm 0.0$ | $0.86 \pm 0.0$ | $0.93 \pm 0.0$ | $0.92 \pm 0.0$ | 1.44(13.39%) | 2.37(-89.09%) | −22225.46(66.83%) | $2.88 \pm 1.11$ |
| COMPAS | GBDT | 0.05 | $0.7 \pm 0.0$ | $0.67 \pm 0.01$ | $0.77 \pm 0.0$ | $0.72 \pm 0.01$ | 1.2(4.35%) | 1.44(-25.77%) | 8.85(8.59%) | $0.77 \pm 0.06$ |
| | | 0.10 | $0.7 \pm 0.0$ | $0.67 \pm 0.01$ | $0.77 \pm 0.0$ | $0.72 \pm 0.01$ | 1.19(3.48%) | 1.4(-27.84%) | 9.11(11.78%) | $0.77 \pm 0.06$ |
| | | 0.20 | $0.7 \pm 0.0$ | $0.67 \pm 0.01$ | $0.77 \pm 0.0$ | $0.72 \pm 0.01$ | 1.12(-2.61%) | 1.3(-32.99%) | 8.97(10.06%) | $0.68 \pm 0.04$ |
| | L1LR | 0.05 | $0.68 \pm 0.0$ | $0.67 \pm 0.01$ | $0.73 \pm 0.0$ | $0.72 \pm 0.01$ | 1.14(-8.8%) | 1.62(-29.87%) | 5.67(11.39%) | $0.29 \pm 0.02$ |
| | | 0.10 | $0.68 \pm 0.0$ | $0.67 \pm 0.01$ | $0.73 \pm 0.0$ | $0.72 \pm 0.01$ | 1.14(-8.8%) | 1.55(-32.9%) | 5.85(14.93%) | $0.29 \pm 0.01$ |
| | | 0.20 | $0.68 \pm 0.0$ | $0.67 \pm 0.01$ | $0.73 \pm 0.0$ | $0.72 \pm 0.01$ | 1.14(-8.8%) | 1.5(-35.06%) | 5.87(15.32%) | $0.28 \pm 0.01$ |
| | L2LR | 0.05 | $0.68 \pm 0.0$ | $0.67 \pm 0.01$ | $0.73 \pm 0.0$ | $0.72 \pm 0.01$ | 1.17(-7.14%) | 1.92(-20.33%) | 6.36(22.54%) | $0.27 \pm 0.01$ |
| | | 0.10 | $0.68 \pm 0.0$ | $0.67 \pm 0.01$ | $0.73 \pm 0.0$ | $0.72 \pm 0.01$ | 1.17(-7.14%) | 1.85(-23.24%) | 6.27(20.81%) | $0.27 \pm 0.01$ |
| | | 0.20 | $0.68 \pm 0.0$ | $0.67 \pm 0.01$ | $0.73 \pm 0.0$ | $0.72 \pm 0.01$ | 1.17(-6.35%) | 1.68(-30.29%) | 6.26(20.62%) | $0.29 \pm 0.01$ |
| | MLP | 0.05 | $0.69 \pm 0.01$ | $0.67 \pm 0.01$ | $0.74 \pm 0.01$ | $0.72 \pm 0.01$ | 1.2(-11.11%) | 1.67(-27.39%) | 8.2(26.35%) | $0.39 \pm 0.07$ |
| | | 0.10 | $0.69 \pm 0.01$ | $0.67 \pm 0.01$ | $0.74 \pm 0.01$ | $0.72 \pm 0.01$ | 1.2(-11.11%) | 1.65(-28.26%) | 8.19(26.19%) | $0.41 \pm 0.06$ |
| | | 0.20 | $0.69 \pm 0.01$ | $0.67 \pm 0.01$ | $0.74 \pm 0.01$ | $0.72 \pm 0.01$ | 1.2(-10.37%) | 1.62(-29.57%) | 8.36(28.81%) | $0.42 \pm 0.07$ |
| Diabetes | GBDT | 0.05 | $0.65 \pm 0.0$ | $0.64 \pm 0.0$ | $0.7 \pm 0.0$ | $0.7 \pm 0.0$ | 1.37(-3.6%) | 1.16(-58.87%) | −4521.05(-92.44%) | $50.03 \pm 8.06$ |
| | | 0.10 | $0.65 \pm 0.0$ | $0.64 \pm 0.0$ | $0.7 \pm 0.0$ | $0.7 \pm 0.0$ | 1.36(-2.16%) | 1.35(-52.13%) | −5505.82(-90.8%) | $58.29 \pm 7.65$ |
| | | 0.20 | $0.65 \pm 0.0$ | $0.64 \pm 0.0$ | $0.7 \pm 0.0$ | $0.7 \pm 0.0$ | 1.35(-2.88%) | 1.46(-48.23%) | −5258.28(-91.21%) | $54.67 \pm 7.11$ |
| | L1LR | 0.05 | $0.62 \pm 0.0$ | $0.62 \pm 0.0$ | $0.66 \pm 0.0$ | $0.66 \pm 0.0$ | 1.2(-25.93%) | 2.31(-11.15%) | −11250.28(46.0%) | $5.23 \pm 0.68$ |
| | | 0.10 | $0.62 \pm 0.0$ | $0.62 \pm 0.0$ | $0.66 \pm 0.0$ | $0.66 \pm 0.0$ | 1.2(-25.93%) | 2.31(-11.15%) | −11190.99(46.29%) | $5.3 \pm 0.7$ |
| | | 0.20 | $0.62 \pm 0.0$ | $0.62 \pm 0.0$ | $0.66 \pm 0.0$ | $0.66 \pm 0.0$ | 1.2(-25.93%) | 2.31(-11.15%) | −7913.34(62.02%) | $5.09 \pm 0.63$ |
| | L2LR | 0.05 | $0.62 \pm 0.0$ | $0.62 \pm 0.0$ | $0.66 \pm 0.0$ | $0.66 \pm 0.0$ | 1.24(-23.46%) | 2.31(-14.44%) | −23047.62(22.58%) | $7.05 \pm 1.0$ |
| | | 0.10 | $0.62 \pm 0.0$ | $0.62 \pm 0.0$ | $0.66 \pm 0.0$ | $0.66 \pm 0.0$ | 1.24(-23.46%) | 2.31(-14.44%) | −23047.64(22.58%) | $7.12 \pm 0.99$ |
| | | 0.20 | $0.62 \pm 0.0$ | $0.62 \pm 0.0$ | $0.66 \pm 0.0$ | $0.66 \pm 0.0$ | 1.24(-23.46%) | 2.31(-14.44%) | −14691.43(21.86%) | $7.41 \pm 0.64$ |
| | MLP | 0.05 | $0.65 \pm 0.01$ | $0.63 \pm 0.0$ | $0.71 \pm 0.01$ | $0.68 \pm 0.0$ | 1.41(-13.5%) | 1.73(-35.45%) | −46675.04(33.81%) | $40.41 \pm 30.18$ |
| | | 0.10 | $0.65 \pm 0.01$ | $0.63 \pm 0.0$ | $0.71 \pm 0.01$ | $0.68 \pm 0.0$ | 1.41(-13.5%) | 1.72(-35.82%) | −46689.47(33.84%) | $38.03 \pm 27.63$ |
| | | 0.20 | $0.65 \pm 0.01$ | $0.63 \pm 0.0$ | $0.71 \pm 0.01$ | $0.68 \pm 0.0$ | 1.39(-14.72%) | 1.73(-35.45%) | −47723.79(4.23%) | $30.72 \pm 19.28$ |
| FICO | GBDT | 0.05 | $0.77 \pm 0.0$ | $0.72 \pm 0.01$ | $0.85 \pm 0.0$ | $0.79 \pm 0.01$ | 1.97(-44.97%) | 0.87(-51.93%) | −58.85(20.61%) | $132.34 \pm 34.38$ |
| | | 0.10 | $0.77 \pm 0.0$ | $0.72 \pm 0.01$ | $0.85 \pm 0.0$ | $0.79 \pm 0.01$ | 2.03(-43.3%) | 0.89(-50.83%) | −58.47(21.13%) | $162.91 \pm 37.45$ |
| | | 0.20 | $0.77 \pm 0.0$ | $0.72 \pm 0.01$ | $0.85 \pm 0.0$ | $0.79 \pm 0.01$ | 2.03(-42.18%) | 0.88(-51.38%) | −56.13(24.28%) | $163.64 \pm 45.55$ |
| | l1lr | 0.05 | $0.71 \pm 0.0$ | $0.7 \pm 0.0$ | $0.78 \pm 0.0$ | $0.77 \pm 0.01$ | 1.84(-25.51%) | 1.89(4.42%) | −77.6(4.56%) | $29.88 \pm 6.18$ |
| | | 0.10 | $0.71 \pm 0.0$ | $0.7 \pm 0.0$ | $0.78 \pm 0.0$ | $0.77 \pm 0.01$ | 1.86(-24.7%) | 1.96(8.29%) | −78.18(3.85%) | $34.15 \pm 7.9$ |
| | | 0.20 | $0.71 \pm 0.0$ | $0.7 \pm 0.0$ | $0.78 \pm 0.0$ | $0.77 \pm 0.01$ | 1.86(-24.7%) | 2.09(15.47%) | −79.92(-1.71%) | $42.69 \pm 9.43$ |
| | L2LR | 0.05 | $0.72 \pm 0.0$ | $0.71 \pm 0.01$ | $0.78 \pm 0.0$ | $0.77 \pm 0.01$ | 2.3(-16.36%) | 1.8(-6.74%) | −57.96(12.02%) | $285.3 \pm 96.59$ |
| | | 0.10 | $0.72 \pm 0.0$ | $0.71 \pm 0.01$ | $0.78 \pm 0.0$ | $0.77 \pm 0.01$ | 2.28(17.09%) | 1.79(-7.25%) | −57.11(10.38%) | $303.19 \pm 98.72$ |
| | | 0.20 | $0.72 \pm 0.0$ | $0.71 \pm 0.01$ | $0.78 \pm 0.0$ | $0.77 \pm 0.01$ | 2.24(-18.55%) | 1.91(-1.04%) | −57.22(10.59%) | $303.85 \pm 97.78$ |
| | MLP | 0.05 | $0.74 \pm 0.01$ | $0.71 \pm 0.01$ | $0.81 \pm 0.01$ | $0.78 \pm 0.01$ | 2.17(-18.11%) | 1.63(-10.93%) | −79.53(15.44%) | $124.03 \pm 50.02$ |
| | | 0.10 | $0.74 \pm 0.01$ | $0.71 \pm 0.01$ | $0.81 \pm 0.01$ | $0.78 \pm 0.01$ | 2.18(-17.74%) | 1.66(-9.29%) | −77.83(12.98%) | $135.6 \pm 56.71$ |
| | | 0.20 | $0.74 \pm 0.01$ | $0.71 \pm 0.01$ | $0.81 \pm 0.01$ | $0.78 \pm 0.01$ | 2.18(-17.74%) | 1.71(-6.56%) | −78.07(13.33%) | $156.08 \pm 70.95$ |
| German Credit | GBDT | 0.05 | $0.96 \pm 0.01$ | $0.75 \pm 0.02$ | $0.99 \pm 0.0$ | $0.77 \pm 0.03$ | 1.21(-12.95%) | 2.13(13.9%) | −31442.17(58.53%) | $6.28 \pm 3.44$ |
| | | 0.10 | $0.96 \pm 0.01$ | $0.75 \pm 0.02$ | $0.99 \pm 0.0$ | $0.77 \pm 0.03$ | 1.21(-12.95%) | 1.8(-3.74%) | −31253.08(58.78%) | $6.87 \pm 3.83$ |
| | | 0.20 | $0.96 \pm 0.01$ | $0.75 \pm 0.02$ | $0.99 \pm 0.0$ | $0.77 \pm 0.03$ | 1.2(-12.23%) | 1.91(2.14%) | −36087.77(52.4%) | $7.78 \pm 4.46$ |
| | L1LR | 0.05 | $0.75 \pm 0.01$ | $0.75 \pm 0.02$ | $0.8 \pm 0.01$ | $0.78 \pm 0.04$ | 1.03(-20.77%) | 2.03(-17.14%) | −24474.67(61.9%) | $0.79 \pm 0.39$ |
| | | 0.10 | $0.75 \pm 0.01$ | $0.75 \pm 0.02$ | $0.8 \pm 0.01$ | $0.77 \pm 0.04$ | 1.04(-20.0%) | 2.01(-17.96%) | −24862.18(-61.3%) | $0.79 \pm 0.38$ |
| | | 0.20 | $0.75 \pm 0.01$ | $0.75 \pm 0.02$ | $0.8 \pm 0.01$ | $0.78 \pm 0.04$ | 1.03(-20.77%) | 2.12(-13.47%) | −25849.27(-59.76%) | $0.7 \pm 0.17$ |
| | L2LR | 0.05 | $0.78 \pm 0.01$ | $0.76 \pm 0.03$ | $0.83 \pm 0.01$ | $0.79 \pm 0.04$ | 1.17(-22.52%) | 3.0(-2.6%) | −40660.55(63.68%) | $2.05 \pm 1.58$ |
| | | 0.10 | $0.78 \pm 0.01$ | $0.76 \pm 0.03$ | $0.83 \pm 0.01$ | $0.79 \pm 0.04$ | 1.18(-21.85%) | 3.03(-1.62%) | −40228.76(64.06%) | $1.84 \pm 1.02$ |
| | | 0.20 | $0.78 \pm 0.01$ | $0.76 \pm 0.03$ | $0.83 \pm 0.01$ | $0.79 \pm 0.04$ | 1.17(-22.52%) | 2.93(-4.87%) | −40136.71(64.15%) | $1.71 \pm 0.82$ |
| | MLP | 0.05 | $0.81 \pm 0.04$ | $0.76 \pm 0.03$ | $0.87 \pm 0.04$ | $0.78 \pm 0.04$ | 1.25(-21.88%) | 2.57(-4.46%) | −46257.34(61.31%) | $2.99 \pm 1.42$ |
| | | 0.10 | $0.81 \pm 0.05$ | $0.76 \pm 0.03$ | $0.87 \pm 0.04$ | $0.78 \pm 0.04$ | 1.23(-23.13%) | 2.56(-4.83%) | −46884.11(60.79%) | $3.04 \pm 1.67$ |
| | | 0.20 | $0.81 \pm 0.04$ | $0.76 \pm 0.03$ | $0.87 \pm 0.04$ | $0.78 \pm 0.04$ | 1.21(-24.38%) | 2.6(-3.35%) | −41223.18(65.52%) | $2.55 \pm 1.47$ |
| Headline | GBDT | 0.05 | $0.82 \pm 0.0$ | $0.81 \pm 0.0$ | $0.9 \pm 0.0$ | $0.89 \pm 0.0$ | 1.74(-4.4%) | 2.49(5.96%) | −407.77(-3.13%) | $22.98 \pm 8.46$ |
| | | 0.10 | $0.82 \pm 0.0$ | $0.81 \pm 0.0$ | $0.9 \pm 0.0$ | $0.89 \pm 0.0$ | 1.71(-6.04%) | 2.51(6.81%) | −432.26(-9.32%) | $20.88 \pm 7.71$ |
| | | 0.20 | $0.82 \pm 0.0$ | $0.81 \pm 0.0$ | $0.9 \pm 0.0$ | $0.89 \pm 0.0$ | 1.53(-15.93%) | 2.22(-5.53%) | −543.65(-37.49%) | $8.83 \pm 2.41$ |
| | L1LR | 0.05 | $0.78 \pm 0.0$ | $0.78 \pm 0.0$ | $0.85 \pm 0.0$ | $0.85 \pm 0.0$ | 1.54(-19.79%) | 2.94(17.13%) | −576.99(-3.25%) | $3.97 \pm 0.15$ |
| | | 0.10 | $0.78 \pm 0.0$ | $0.78 \pm 0.0$ | $0.85 \pm 0.0$ | $0.85 \pm 0.0$ | 1.55(-19.27%) | 2.94(17.13%) | −577.03(-3.26%) | $4.16 \pm 0.17$ |
| | | 0.20 | $0.78 \pm 0.0$ | $0.78 \pm 0.0$ | $0.85 \pm 0.0$ | $0.85 \pm 0.0$ | 1.47(-23.44%) | 2.94(17.13%) | −577.7(-3.38%) | $2.54 \pm 0.12$ |
| | L2LR | 0.05 | $0.78 \pm 0.0$ | $0.78 \pm 0.0$ | $0.86 \pm 0.0$ | $0.85 \pm 0.0$ | 1.59(-19.7%) | 2.94(-17.6%) | −556.65(0.13%) | $4.81 \pm 0.2$ |
| | | 0.10 | $0.78 \pm 0.0$ | $0.78 \pm 0.0$ | $0.85 \pm 0.0$ | $0.85 \pm 0.0$ | 1.6(-19.19%) | 2.94(17.6%) | −573.97(-3.24%) | $5.1 \pm 0.25$ |
| | | 0.20 | $0.78 \pm 0.0$ | $0.78 \pm 0.0$ | $0.85 \pm 0.0$ | $0.85 \pm 0.0$ | 1.5(-24.24%) | 2.94(17.6%) | −574.67(-3.37%) | $3.22 \pm 0.13$ |
| | MLP | 0.05 | $0.83 \pm 0.01$ | $0.81 \pm 0.0$ | $0.91 \pm 0.01$ | $0.89 \pm 0.0$ | 1.64(-19.21%) | 1.97(-14.72%) | −617.43(-25.15%) | $7.02 \pm 1.86$ |
| | | 0.10 | $0.83 \pm 0.01$ | $0.81 \pm 0.0$ | $0.91 \pm 0.01$ | $0.89 \pm 0.0$ | 1.64(-19.21%) | 1.97(-14.72%) | −604.44(-22.51%) | $7.47 \pm 2.23$ |
| | | 0.20 | $0.83 \pm 0.01$ | $0.81 \pm 0.0$ | $0.91 \pm 0.01$ | $0.89 \pm 0.0$ | 1.5(-26.11%) | 2.06(-10.82%) | −570.13(-15.56%) | $4.1 \pm 0.79$ |
| MIMIC | GBDT | 0.05 | $0.91 \pm 0.0$ | $0.9 \pm 0.0$ | $0.87 \pm 0.0$ | $0.85 \pm 0.0$ | 1.21(2.54%) | 0.52(-59.38%) | −19.06(-0.74%) | $2.93 \pm 0.39$ |
| | | 0.10 | $0.91 \pm 0.0$ | $0.9 \pm 0.0$ | $0.87 \pm 0.0$ | $0.85 \pm 0.0$ | 1.21(2.54%) | 0.48(-62.5%) | −19.08(-0.85%) | $2.98 \pm 0.39$ |
| | | 0.20 | $0.91 \pm 0.0$ | $0.9 \pm 0.0$ | $0.87 \pm 0.0$ | $0.85 \pm 0.0$ | 1.21(2.54%) | 0.41(-67.97%) | −18.86(0.32%) | $3.32 \pm 0.43$ |
| | L1LR | 0.05 | $0.89 \pm 0.0$ | $0.89 \pm 0.0$ | $0.8 \pm 0.0$ | $0.8 \pm 0.0$ | 1.17(1.74%) | 1.11(-75.5%) | −21.32(-7.89%) | $0.75 \pm 0.06$ |
| | | 0.10 | $0.89 \pm 0.0$ | $0.89 \pm 0.0$ | $0.8 \pm 0.0$ | $0.8 \pm 0.0$ | 1.18(2.61%) | 1.15(-74.61%) | −21.48(-8.7%) | $0.77 \pm 0.07$ |
| | | 0.20 | $0.89 \pm 0.0$ | $0.89 \pm 0.0$ | $0.8 \pm 0.0$ | $0.8 \pm 0.0$ | 1.18(2.61%) | 1.15(-74.61%) | −21.48(-8.7%) | $0.79 \pm 0.08$ |
| | L2LR | 0.05 | $0.89 \pm 0.0$ | $0.89 \pm 0.0$ | $0.8 \pm 0.0$ | $0.8 \pm 0.0$ | 1.19(2.59%) | 1.15(-73.5%) | −21.37(-8.7%) | $0.86 \pm 0.1$ |
| | | 0.10 | $0.89 \pm 0.0$ | $0.89 \pm 0.0$ | $0.8 \pm 0.0$ | $0.8 \pm 0.0$ | 1.19(2.59%) | 1.15(-73.5%) | −21.41(-8.9%) | $0.84 \pm 0.09$ |
| | | 0.20 | $0.89 \pm 0.0$ | $0.89 \pm 0.0$ | $0.8 \pm 0.0$ | $0.8 \pm 0.0$ | 1.19(2.59%) | 1.15(-73.5%) | −21.48(-9.26%) | $0.91 \pm 0.09$ |
| | MLP | 0.05 | $0.9 \pm 0.0$ | $0.9 \pm 0.0$ | $0.87 \pm 0.01$ | $0.85 \pm 0.0$ | 1.21(2.54%) | 0.58(-72.12%) | −18.22(-5.62%) | $1.35 \pm 0.15$ |
| | | 0.10 | $0.9 \pm 0.0$ | $0.9 \pm 0.0$ | $0.87 \pm 0.01$ | $0.85 \pm 0.0$ | 1.22(3.39%) | 0.58(-72.12%) | −18.12(-5.04%) | $1.41 \pm 0.14$ |
| | | 0.20 | $0.9 \pm 0.0$ | $0.9 \pm 0.0$ | $0.87 \pm 0.01$ | $0.85 \pm 0.0$ | 1.22(3.39%) | 0.58(-72.12%) | −18.12(-5.04%) | $1.43 \pm 0.14$ |

# I   All-Opt$^-$ variants performance

In this section, we will mainly show the model performance of All-Opt$^©$ and All-Opt$^1$, which are the two gradient-based optimization methods used for SEV$^©$ and SEV$^1$ optimization. Table 10 shows the SEV$^1$, $\ell_\infty$ and model performance after applying All-Opt$^1$ methods for different models on different datasets with different levels of flexibility. It is evident that All-Opt$^F$ has provided a significant decrease in SEV, so that its values are close to 1, providing much sparser explanations without model performance loss and closeness/credibility loss in explanations. Similar findings are observed in Table 11.

Table 10: The model performance for All-Opt$^1$

| Dataset | Model | Train Accuracy | Test Accuracy | Train AUC | Test AUC | Mean SEV$^-$ | Mean $\ell_\infty$ | Training Time(s) | Mean Log-likelihood |
|---|---|---|---|---|---|---|---|---|---|
| Adult | GBDT | $0.87 \pm 0.02$ | $0.84 \pm 0.02$ | $0.93 \pm 0.01$ | $0.90 \pm 0.01$ | $1.00 \pm 0.00$ | $5.67 \pm 0.34$ | $2010 \pm 24$ | $-39654.89 \pm 4201.17$ |
| | LR | $0.84 \pm 0.01$ | $0.84 \pm 0.01$ | $0.90 \pm 0.02$ | $0.89 \pm 0.01$ | $1.03 \pm 0.01$ | $3.21 \pm 0.02$ | $60 \pm 1$ | $-70566.06 \pm 10678.32$ |
| | MLP | $0.86 \pm 0.01$ | $0.85 \pm 0.01$ | $0.91 \pm 0.02$ | $0.91 \pm 0.01$ | $1.00 \pm 0.00$ | $9.52 \pm 1.45$ | $82 \pm 3$ | $-58049.77 \pm 9932.16$ |
| COMPAS | GBDT | $0.70 \pm 0.01$ | $0.68 \pm 0.01$ | $0.74 \pm 0.01$ | $0.71 \pm 0.01$ | $1.01 \pm 0.01$ | $1.50 \pm 0.04$ | $244 \pm 4$ | $10.74 \pm 0.98$ |
| | LR | $0.68 \pm 0.01$ | $0.68 \pm 0.02$ | $0.74 \pm 0.01$ | $0.73 \pm 0.02$ | $1.00 \pm 0.00$ | $2.13 \pm 0.01$ | $11 \pm 1$ | $9.17 \pm 1.02$ |
| | MLP | $0.68 \pm 0.01$ | $0.67 \pm 0.02$ | $0.74 \pm 0.02$ | $0.72 \pm 0.01$ | $1.01 \pm 0.01$ | $1.90 \pm 0.11$ | $16 \pm 1$ | $14.57 \pm 1.23$ |
| Diabetes | GBDT | $0.62 \pm 0.01$ | $0.63 \pm 0.01$ | $0.62 \pm 0.01$ | $0.64 \pm 0.01$ | $1.07 \pm 0.01$ | $1.78 \pm 0.34$ | $10548 \pm 324$ | $-14013.49 \pm 2784.36$ |
| | LR | $0.62 \pm 0.04$ | $0.62 \pm 0.04$ | $0.63 \pm 0.01$ | $0.63 \pm 0.01$ | $1.07 \pm 0.00$ | $1.39 \pm 0.05$ | $217 \pm 3$ | $-40190.09 \pm 10453.69$ |
| | MLP | $0.62 \pm 0.01$ | $0.65 \pm 0.01$ | $0.65 \pm 0.01$ | $0.64 \pm 0.02$ | $1.07 \pm 0.00$ | $2.50 \pm 0.32$ | $318 \pm 5$ | $-18013.49 \pm 3894.36$ |
| FICO | GBDT | $0.70 \pm 0.02$ | $0.70 \pm 0.02$ | $0.77 \pm 0.01$ | $0.77 \pm 0.02$ | $1.19 \pm 0.10$ | $0.84 \pm 0.12$ | $864 \pm 23$ | $-40.44 \pm 4.32$ |
| | LR | $0.70 \pm 0.02$ | $0.70 \pm 0.02$ | $0.77 \pm 0.01$ | $0.77 \pm 0.02$ | $1.10 \pm 0.10$ | $1.91 \pm 0.33$ | $19 \pm 1$ | $-20.32 \pm 0.18$ |
| | MLP | $0.72 \pm 0.01$ | $0.72 \pm 0.01$ | $0.78 \pm 0.02$ | $0.78 \pm 0.01$ | $1.28 \pm 0.09$ | $1.23 \pm 0.21$ | $28 \pm 0$ | $-26.04 \pm 0.43$ |
| German Credit | GBDT | $0.94 \pm 0.02$ | $0.73 \pm 0.02$ | $0.99 \pm 0.01$ | $0.76 \pm 0.02$ | $1.02 \pm 0.01$ | $1.21 \pm 0.05$ | $99 \pm 1$ | $-27701.04 \pm 3431.99$ |
| | LR | $0.77 \pm 0.01$ | $0.75 \pm 0.01$ | $0.82 \pm 0.02$ | $0.77 \pm 0.01$ | $1.00 \pm 0.00$ | $1.39 \pm 0.05$ | $2 \pm 0$ | $-58065.80 \pm 6843.21$ |
| | MLP | $0.82 \pm 0.01$ | $0.73 \pm 0.03$ | $0.93 \pm 0.02$ | $0.75 \pm 0.02$ | $1.00 \pm 0.00$ | $1.17 \pm 0.08$ | $3 \pm 1$ | $-85816.95 \pm 13728.23$ |
| Headline | GBDT | $0.80 \pm 0.01$ | $0.76 \pm 0.02$ | $0.90 \pm 0.01$ | $0.89 \pm 0.01$ | $1.04 \pm 0.02$ | $2.45 \pm 0.57$ | $2732 \pm 101$ | $-4.37 \pm 1.28$ |
| | LR | $0.77 \pm 0.01$ | $0.78 \pm 0.01$ | $0.86 \pm 0.01$ | $0.85 \pm 0.01$ | $1.00 \pm 0.00$ | $2.77 \pm 0.44$ | $78 \pm 0$ | $-2.39 \pm 0.11$ |
| | MLP | $0.76 \pm 0.02$ | $0.77 \pm 0.03$ | $0.87 \pm 0.02$ | $0.86 \pm 0.02$ | $1.03 \pm 0.03$ | $2.78 \pm 0.13$ | $102 \pm 1$ | $-2.57 \pm 0.89$ |
| MIMIC | GBDT | $0.88 \pm 0.01$ | $0.88 \pm 0.01$ | $0.84 \pm 0.01$ | $0.82 \pm 0.01$ | $1.06 \pm 0.04$ | $3.66 \pm 0.02$ | $2799 \pm 102$ | $-16.36 \pm 0.54$ |
| | LR | $0.88 \pm 0.01$ | $0.88 \pm 0.01$ | $0.84 \pm 0.01$ | $0.82 \pm 0.02$ | $1.03 \pm 0.03$ | $3.67 \pm 0.72$ | $87 \pm 2$ | $-17.77 \pm 2.22$ |
| | MLP | $0.89 \pm 0.01$ | $0.89 \pm 0.02$ | $0.84 \pm 0.03$ | $0.82 \pm 0.03$ | $1.00 \pm 0.00$ | $1.29 \pm 0.20$ | $115 \pm 2$ | $-10.38 \pm 3.87$ |

Table 11: The model performance for All-Opt$^©$

| Dataset | Model | Train Accuracy | Test Accuracy | Train AUC | Test AUC | Mean SEV$^©$ | Mean $\ell_\infty$ | Mean Log-likelihood |
|---|---|---|---|---|---|---|---|---|
| Adult | GBDT | $0.90 \pm 0.00$ | $0.83 \pm 0.01$ | $0.89 \pm 0.01$ | $0.89 \pm 0.01$ | $1.14 \pm 0.03$ | $1.87 \pm 0.03$ | $289.07 \pm 52.79$ |
| | LR | $0.84 \pm 0.00$ | $0.84 \pm 0.01$ | $0.91 \pm 0.01$ | $0.90 \pm 0.01$ | $1.01 \pm 0.01$ | $2.56 \pm 0.43$ | $299.04 \pm 17.24$ |
| | MLP | $0.85 \pm 0.01$ | $0.84 \pm 0.01$ | $0.92 \pm 0.01$ | $0.91 \pm 0.01$ | $1.00 \pm 0.00$ | $2.37 \pm 0.19$ | $297.14 \pm 32.16$ |
| COMPAS | GBDT | $0.68 \pm 0.01$ | $0.68 \pm 0.01$ | $0.72 \pm 0.01$ | $0.74 \pm 0.02$ | $1.02 \pm 0.02$ | $1.34 \pm 0.47$ | $10.28 \pm 2.14$ |
| | LR | $0.68 \pm 0.01$ | $0.68 \pm 0.01$ | $0.72 \pm 0.01$ | $0.74 \pm 0.02$ | $1.00 \pm 0.00$ | $2.49 \pm 0.21$ | $8.67 \pm 1.32$ |
| | MLP | $0.67 \pm 0.01$ | $0.67 \pm 0.02$ | $0.72 \pm 0.01$ | $0.74 \pm 0.01$ | $1.05 \pm 0.05$ | $1.92 \pm 0.05$ | $7.22 \pm 0.56$ |
| Diabetes | GBDT | $0.62 \pm 0.01$ | $0.62 \pm 0.02$ | $0.66 \pm 0.01$ | $0.66 \pm 0.02$ | $1.05 \pm 0.00$ | $1.99 \pm 0.01$ | $-5231.53 \pm 489.52$ |
| | LR | $0.62 \pm 0.01$ | $0.62 \pm 0.02$ | $0.66 \pm 0.01$ | $0.66 \pm 0.02$ | $1.05 \pm 0.00$ | $2.89 \pm 0.46$ | $-5937.66 \pm 638.77$ |
| | MLP | $0.62 \pm 0.01$ | $0.62 \pm 0.01$ | $0.67 \pm 0.01$ | $0.67 \pm 0.01$ | $1.05 \pm 0.00$ | $2.12 \pm 0.01$ | $-5217.39 \pm 497.78$ |
| FICO | GBDT | $0.70 \pm 0.01$ | $0.70 \pm 0.00$ | $0.78 \pm 0.01$ | $0.78 \pm 0.01$ | $1.48 \pm 0.09$ | $0.90 \pm 0.01$ | $-55.09 \pm 6.79$ |
| | LR | $0.70 \pm 0.01$ | $0.70 \pm 0.00$ | $0.78 \pm 0.01$ | $0.78 \pm 0.01$ | $1.41 \pm 0.08$ | $1.60 \pm 0.27$ | $-15.66 \pm 7.01$ |
| | MLP | $0.70 \pm 0.01$ | $0.69 \pm 0.11$ | $0.79 \pm 0.02$ | $0.78 \pm 0.02$ | $1.28 \pm 0.19$ | $1.23 \pm 0.05$ | $-18.47 \pm 8.98$ |
| German Credit | GBDT | $0.75 \pm 0.01$ | $0.76 \pm 0.01$ | $0.82 \pm 0.01$ | $0.80 \pm 0.01$ | $1.00 \pm 0.00$ | $1.00 \pm 0.00$ | $-15797.31 \pm 2134.01$ |
| | LR | $0.75 \pm 0.01$ | $0.76 \pm 0.01$ | $0.82 \pm 0.01$ | $0.80 \pm 0.01$ | $1.00 \pm 0.00$ | $1.00 \pm 0.00$ | $-45070.76 \pm 7924.23$ |
| | MLP | $0.86 \pm 0.02$ | $0.79 \pm 0.01$ | $0.92 \pm 0.01$ | $0.80 \pm 0.01$ | $1.00 \pm 0.00$ | $1.00 \pm 0.00$ | $-30917.95 \pm 5534.23$ |
| Headline | GBDT | $0.78 \pm 0.02$ | $0.79 \pm 0.01$ | $0.85 \pm 0.01$ | $0.85 \pm 0.01$ | $1.26 \pm 0.03$ | $-1.72 \pm 0.01$ | $-4.20 \pm 2.97$ |
| | LR | $0.78 \pm 0.02$ | $0.79 \pm 0.01$ | $0.85 \pm 0.01$ | $0.85 \pm 0.01$ | $1.29 \pm 0.10$ | $2.93 \pm 0.02$ | $-2.93 \pm 1.28$ |
| | MLP | $0.78 \pm 0.02$ | $0.78 \pm 0.03$ | $0.84 \pm 0.01$ | $0.84 \pm 0.01$ | $1.15 \pm 0.12$ | $1.69 \pm 0.16$ | $-2.87 \pm 1.51$ |
| MIMIC | GBDT | $0.90 \pm 0.01$ | $0.89 \pm 0.01$ | $0.80 \pm 0.00$ | $0.80 \pm 0.00$ | $1.05 \pm 0.05$ | $1.00 \pm 0.00$ | $-21.80 \pm 2.45$ |
| | LR | $0.90 \pm 0.01$ | $0.89 \pm 0.01$ | $0.80 \pm 0.00$ | $0.80 \pm 0.00$ | $1.00 \pm 0.00$ | $1.00 \pm 0.00$ | $-28.74 \pm 0.75$ |
| | MLP | $0.89 \pm 0.01$ | $0.89 \pm 0.01$ | $0.84 \pm 0.01$ | $0.81 \pm 0.00$ | $1.01 \pm 0.01$ | $0.06 \pm 0.01$ | $-29.35 \pm 0.36$ |

# J   $SEV^T$ in tree-based models

In this section, we show the model performance and $SEV^T$ values for different types of tree-based models. As discussed in section 4.2, the similarity and closeness metrics in $SEV^T$ are all $\ell_0$ norm, so we only need to compute the mean $SEV^T$ for each tree. Table 12 shows that most of the tree-based models can provide sparse explanations ($SEV^T \leq 2$), and we can also find a decision tree with the same model performance as the other tree-based models from $SEV^T$=1 to TOpt.

Table 12: The model performance with different tree-based methods

| DATASET | METHODS | TRAIN ACC | TEST ACC | MEAN $SEV^T$ |
|---------|---------|-----------|----------|--------------|
| Adult | CART | $0.84 \pm 0.01$ | $0.84 \pm 0.01$ | $1.11 \pm 0.01$ |
| | C4.5 | $0.85 \pm 0.01$ | $0.84 \pm 0.00$ | $1.10 \pm 0.02$ |
| | GOSDT | $0.81 \pm 0.01$ | $0.81 \pm 0.01$ | $1.08 \pm 0.01$ |
| | Topt | $0.82 \pm 0.01$ | $0.82 \pm 0.01$ | $1.00 \pm 0.00$ |
| COMPAS | CART | $0.68 \pm 0.00$ | $0.65 \pm 0.01$ | $1.02 \pm 0.01$ |
| | C4.5 | $0.68 \pm 0.00$ | $0.65 \pm 0.01$ | $1.02 \pm 0.01$ |
| | GOSDT | $0.67 \pm 0.02$ | $0.65 \pm 0.01$ | $1.12 \pm 0.02$ |
| | Topt | $0.66 \pm 0.01$ | $0.67 \pm 0.01$ | $1.00 \pm 0.00$ |
| Diabetes | CART | $0.63 \pm 0.01$ | $0.63 \pm 0.01$ | $1.00 \pm 0.00$ |
| | C4.5 | $0.63 \pm 0.01$ | $0.63 \pm 0.01$ | $1.00 \pm 0.00$ |
| | GOSDT | $0.61 \pm 0.01$ | $0.60 \pm 0.01$ | $1.00 \pm 0.00$ |
| | Topt | $0.62 \pm 0.01$ | $0.63 \pm 0.01$ | $1.00 \pm 0.00$ |
| FICO | CART | $0.71 \pm 0.01$ | $0.71 \pm 0.01$ | $1.10 \pm 0.03$ |
| | C4.5 | $0.71 \pm 0.01$ | $0.71 \pm 0.01$ | $1.13 \pm 0.05$ |
| | GOSDT | $0.70 \pm 0.01$ | $0.69 \pm 0.01$ | $1.80 \pm 0.02$ |
| | Topt | $0.70 \pm 0.01$ | $0.71 \pm 0.01$ | $1.00 \pm 0.00$ |
| German | CART | $0.75 \pm 0.01$ | $0.70 \pm 0.01$ | $1.00 \pm 0.00$ |
| Credit | C4.5 | $0.75 \pm 0.01$ | $0.70 \pm 0.01$ | $1.00 \pm 0.00$ |
| | GOSDT | $0.75 \pm 0.01$ | $0.70 \pm 0.01$ | $1.00 \pm 0.00$ |
| | Topt | $0.75 \pm 0.01$ | $0.70 \pm 0.01$ | $1.00 \pm 0.00$ |
| Headline | CART | $0.78 \pm 0.01$ | $0.78 \pm 0.00$ | $1.27 \pm 0.01$ |
| | C4.5 | $0.77 \pm 0.01$ | $0.77 \pm 0.00$ | $1.16 \pm 0.02$ |
| | GOSDT | $0.76 \pm 0.01$ | $0.76 \pm 0.02$ | $1.09 \pm 0.02$ |
| | Topt | $0.77 \pm 0.00$ | $0.77 \pm 0.00$ | $1.00 \pm 0.00$ |
| MIMIC | CART | $0.89 \pm 0.01$ | $0.89 \pm 0.01$ | $1.00 \pm 0.00$ |
| | C4.5 | $0.89 \pm 0.01$ | $0.89 \pm 0.01$ | $1.00 \pm 0.00$ |
| | GOSDT | $0.89 \pm 0.01$ | $0.89 \pm 0.01$ | $1.00 \pm 0.00$ |
| | Topt | $0.89 \pm 0.01$ | $0.89 \pm 0.01$ | $1.00 \pm 0.00$ |

# K    The SEV[1] results after ExpO optimization

For the ExpO comparison experiment, we used the fidelity metrics from Plumb et al. [2020] as the penalty term for regularizing the original model. Then we evaluated the optimized model with $SEV^-$. We used two kinds of fidelity metrics as the regularization term: 1D fidelity and 1D fidelity. Both of these two penalty terms aim to optimize the model $f$ such that the local model $g$ [Ribeiro et al., 2016, Plumb et al., 2018] accurately approximates $f$ in the neighborhood $N_x$, which is equivalent to minimizing:

$$\ell_{\text{fed}}(f, g, N_x) = \mathbb{E}_{\boldsymbol{x}' \sim N_{\boldsymbol{x}}}[g(\boldsymbol{x}') - f(\boldsymbol{x}')]^2. \tag{10}$$

The local model $g$'s are linear models, and the $N_{\boldsymbol{x}}$ are points sampled normally around the original query. The 1D version of Fidelity regularization requires sampling the points around each feature of $\boldsymbol{x}$ at a time, which saves time and computational complexity. Based on the above equation, we rewrite the overall objective function as:

$$\min_{f \in \mathcal{F}} \ell_{\text{BCE}} + C_F \ell_{\text{fed}} \tag{11}$$

where $\ell_{\text{BCE}}$ is the Binary Cross Entropy Loss to control the accuracy of the training model, $C_F$ is the strength of the fidelity term, and the training process is the same All-Opt$^-$ optimization, which we used 80 epochs for basic training process, 20 epochs for regularization.

In this section, we show the $SEV^-$ and training time for ExpO regularizer in **LR** and **MLP** models with 1D Fidelity (1DFed) and Global Fidelity (Fed) regularizers. Comparing the mean $SEV^1$ of Table 13 with Table 7, it is evident that with the optimization through Fed or 1DFed, the optimized models do not provide sparse explanations. In addition, it takes a long time to calculate Fed and 1DFed since the regularizer's complexity is determined by the number of queries, features, as well as the points samples around the queries. For $SEV^-$, the complexity is determined only by the number of queries and the number of features, so it is much easier to calculate.

Table 13: Model performance, $SEV^1$ and training time of LR and MLPs after ExpO with different datasets

| DATASET | MODEL | REGULARIZER | TRAIN ACCURACY | TEST ACCURACY | TRAIN AUC | TEST AUC | MEAN SEV$^1$ | TRAINING TIME(S) |
|---|---|---|---|---|---|---|---|---|
| Adult | LR | Fed | $0.85 \pm 0.01$ | $0.84 \pm 0.01$ | $0.90 \pm 0.01$ | $0.89 \pm 0.01$ | $1.23 \pm 0.02$ | $1350 \pm 162$ |
| | | 1DFed | $0.84 \pm 0.02$ | $0.84 \pm 0.01$ | $0.90 \pm 0.01$ | $0.90 \pm 0.02$ | $1.17 \pm 0.02$ | $510 \pm 23$ |
| | MLP | Fed | $0.85 \pm 0.01$ | $0.83 \pm 0.02$ | $0.90 \pm 0.01$ | $0.89 \pm 0.01$ | $1.27 \pm 0.02$ | $1580 \pm 50$ |
| | | 1DFed | $0.85 \pm 0.01$ | $0.83 \pm 0.02$ | $0.90 \pm 0.01$ | $0.89 \pm 0.01$ | $1.27 \pm 0.02$ | $686 \pm 23$ |
| COMPAS | LR | Fed | $0.67 \pm 0.02$ | $0.66 \pm 0.01$ | $0.72 \pm 0.02$ | $0.72 \pm 0.02$ | $1.22 \pm 0.04$ | $58 \pm 10$ |
| | | 1DFed | $0.65 \pm 0.02$ | $0.65 \pm 0.01$ | $0.73 \pm 0.01$ | $0.72 \pm 0.02$ | $1.27 \pm 0.02$ | $90 \pm 5$ |
| | MLP | Fed | $0.68 \pm 0.02$ | $0.66 \pm 0.01$ | $0.74 \pm 0.02$ | $0.72 \pm 0.01$ | $1.28 \pm 0.03$ | $125 \pm 14$ |
| | | 1DFed | $0.66 \pm 0.02$ | $0.66 \pm 0.02$ | $0.72 \pm 0.02$ | $0.71 \pm 0.01$ | $1.28 \pm 0.2$ | $128 \pm 15$ |
| Diabetes | LR | Fed | $0.63 \pm 0.02$ | $0.62 \pm 0.01$ | $0.60 \pm 0.02$ | $0.60 \pm 0.01$ | $1.50 \pm 0.01$ | $3625 \pm 412$ |
| | | 1DFed | $0.63 \pm 0.02$ | $0.62 \pm 0.01$ | $0.60 \pm 0.02$ | $0.60 \pm 0.01$ | $1.46 \pm 0.01$ | $1842 \pm 245$ |
| | MLP | Fed | $0.63 \pm 0.02$ | $0.62 \pm 0.01$ | $0.60 \pm 0.02$ | $0.60 \pm 0.01$ | $1.52 \pm 0.01$ | $4372 \pm 316$ |
| | | 1DFed | $0.63 \pm 0.02$ | $0.62 \pm 0.01$ | $0.60 \pm 0.02$ | $0.60 \pm 0.01$ | $1.46 \pm 0.01$ | $2032 \pm 124$ |
| FICO | LR | Fed | $0.71 \pm 0.01$ | $0.71 \pm 0.01$ | $0.78 \pm 0.02$ | $0.78 \pm 0.01$ | $2.76 \pm 0.12$ | $150 \pm 21$ |
| | | 1DFed | $0.71 \pm 0.02$ | $0.71 \pm 0.01$ | $0.77 \pm 0.01$ | $0.78 \pm 0.01$ | $2.76 \pm 0.21$ | $150 \pm 14$ |
| | MLP | Fed | $0.72 \pm 0.02$ | $0.71 \pm 0.01$ | $0.79 \pm 0.02$ | $0.78 \pm 0.02$ | $2.67 \pm 0.14$ | $210 \pm 13$ |
| | | 1DFed | $0.72 \pm 0.02$ | $0.71 \pm 0.01$ | $0.78 \pm 0.02$ | $0.77 \pm 0.01$ | $2.80 \pm 0.35$ | $195 \pm 14$ |
| German Credit | LR | Fed | $0.78 \pm 0.02$ | $0.76 \pm 0.01$ | $0.82 \pm 0.02$ | $0.80 \pm 0.01$ | $1.65 \pm 0.12$ | $28 \pm 0$ |
| | | 1DFed | $0.77 \pm 0.02$ | $0.73 \pm 0.02$ | $0.80 \pm 0.01$ | $0.76 \pm 0.02$ | $1.76 \pm 0.02$ | $15 \pm 0$ |
| | MLP | Fed | $0.75 \pm 0.02$ | $0.72 \pm 0.02$ | $0.82 \pm 0.01$ | $0.78 \pm 0.02$ | $1.70 \pm 0.03$ | $33 \pm 2$ |
| | | 1DFed | $0.70 \pm 0.00$ | $0.70 \pm 0.00$ | $0.72 \pm 0.02$ | $0.73 \pm 0.01$ | $1.70 \pm 0.03$ | $20 \pm 0$ |
| Headline | LR | Fed | $0.77 \pm 0.04$ | $0.77 \pm 0.01$ | $0.85 \pm 0.01$ | $0.85 \pm 0.00$ | $1.87 \pm 0.01$ | $680 \pm 21$ |
| | | 1DFed | $0.77 \pm 0.01$ | $0.77 \pm 0.01$ | $0.84 \pm 0.01$ | $0.85 \pm 0.01$ | $1.87 \pm 0.02$ | $562 \pm 32$ |
| | MLP | Fed | $0.77 \pm 0.02$ | $0.78 \pm 0.01$ | $0.85 \pm 0.02$ | $0.85 \pm 0.03$ | $1.87 \pm 0.04$ | $762 \pm 56$ |
| | | 1DFed | $0.77 \pm 0.02$ | $0.77 \pm 0.01$ | $0.85 \pm 0.02$ | $0.85 \pm 0.01$ | $1.87 \pm 0.04$ | $852 \pm 72$ |
| MIMIC | LR | Fed | $0.89 \pm 0.02$ | $0.89 \pm 0.02$ | $0.77 \pm 0.01$ | $0.77 \pm 0.01$ | $1.18 \pm 0.02$ | $712 \pm 42$ |
| | | 1DFed | $0.89 \pm 0.02$ | $0.88 \pm 0.01$ | $0.78 \pm 0.02$ | $0.77 \pm 0.02$ | $1.17 \pm 0.02$ | $646 \pm 42$ |
| | MLP | Fed | $0.88 \pm 0.00$ | $0.88 \pm 0.00$ | $0.78 \pm 0.00$ | $0.77 \pm 0.01$ | $1.15 \pm 0.01$ | $960 \pm 27$ |
| | | 1DFed | $0.88 \pm 0.01$ | $0.88 \pm 0.01$ | $0.78 \pm 0.01$ | $0.78 \pm 0.01$ | $1.16 \pm 0.01$ | $873 \pm 18$ |

# L   Proof of Theorem 4.1

**Theorem L.1.** *With a single decision classifier DT and a positively-predicted query $x_i$, define $N_i$ as the leaf that captures it. If $N_i$ has a sibling leaf, or any internal node in its decision path has a negatively-predicted child leaf, then $SEV^T$ is **equal to 1**.*

$SEV^-$ is defined as the number of features that need to change within the given classification tree. If you have switched a particular node from one path to another, it adds one to $SEV^-$. Therefore, for the internal nodes along the $SEV^-$ path, if $N_i$ has a sibling leaf node, if we goes up to its parent node and goes the opposite direction to change the query value for counterfactual explanation, the modified instance will be directly predicted as negative, which leads to $SEV^-$ being equal to 1 in this case.

Figure 11 shows an example for $SEV^T$ being exactly 1, and a case illustrating that if $N$ does not have a sibling or any internal node in its decision path that has a negatively-predicted child leaf, $SEV^T$ should be greater than or equal to 1. In Figure 11, the left trees are the full decision trees, where the blue nodes are the negatively predicted leaf nodes and the red ones are positively predicted. The red arrows graph represents the decision path for a specific instance. The person icon with a plus sign is $N_i$ that we would like to calculate $SEV^T$ on. The right tree is the subtree of the left tree. The person icon with a minus is the query and the blue arrows indicate a decision pathway for SEV Explanation.

If the query is predicted as positive in node Ⓐ, it is easy to see that if we go up to node Ⓒ and goes the opposite direction as the decision path for $x_i$, then you can directly get a negative prediction. In other words, if you change the feature $C$ in the query to make it doens't satisfy the node Ⓒ's condition, then it can be prediction as negative, which means that $SEV^T=1$.

For $SEV^T \geq 1$ case, if the query predidcted as positive in node ⑦, since it does not have a sibling leaf node, then if it goes to its parent node Ⓓ and goes the opposite direction, then it would reach node Ⓔ. However, if we don't know the query $x_i$'s value, then I am unable to know whether I need to change the condition in node Ⓔ for higher $SEV^T$. Therefore, in this case $SEV^T$ can be only guaranteed to be greater or equal to 1.

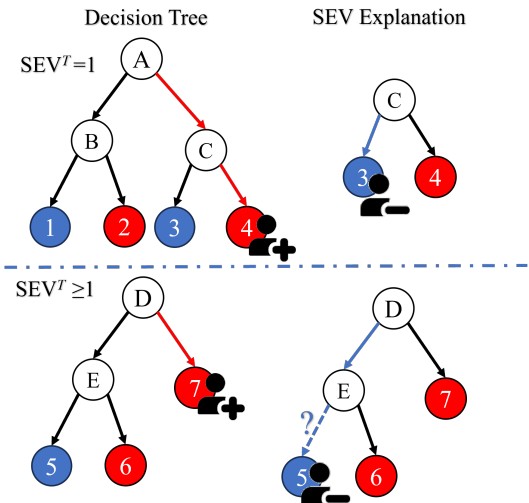

Figure 11: Example of $SEV^T=1$ in Theorem 4.1

# M  Proof of Theorem 4.2

**Theorem M.1.** *With a single decision tree classifier $DT$ and a positively-predicted query $x_i$, with the set of all negatively predicted leaves as reference points, both $SEV^-$ and the $\ell_0$ distance (edit distance) between the query and the $SEV^-$ explanation is minimized.*

**Proof (Optimality of Explanation Path):**

The definition for $SEV^-$ is the minimum number of features that is needed for a positively predicted query $x_i$ to aligned with the reference point in order to be predicted as negative. For tree-based classifiers, the decisions are all made in the leaf nodes. Since we have set of all the negatively predicted leaves as the reference points, then the $\ell_0$ distance (edit distance) between the query and the $SEV^-$ explanation is equivalent to be the minimum $\ell_0$ distance between the query and the negatively predicted leaf nodes. Each node can be considered as a list of rules of conditions that needs to be satisfied. If a query would like to be predicted as negative in a specific node, then it needs to change some of the feature values in the query so as to be predicted as negative, and the number of changed feature is $SEV^-$. Therefore, $SEV^-$ and the $\ell_0$ distance are the same in this theorem.

Next, we would like to show that if one of the negatively predicted leaf nodes is not considered as reference point, then $SEV^-$ is not minimized. It is really easy to give an counterexample: if we have a decision tree shown in Figure 12 with white nodes as root/internal nodes, blue nodes as negatively predicted node, and the red ones as positively predicted. Suppose we have a query predicted as positive, with feature values $\{A : \text{False}, B : \text{False}, C : \text{False}\}$, and only regard node ① as the reference point, then both feature $A$ and $C$ should be change to True, in order to do a negative prediction, in other words, if only node ① is the reference point, then $SEV^-=2$. However, based on Theorem 4.1, since node ④ has a sibling leaf predicted as negative, then the $SEV^-$ is not minimized.

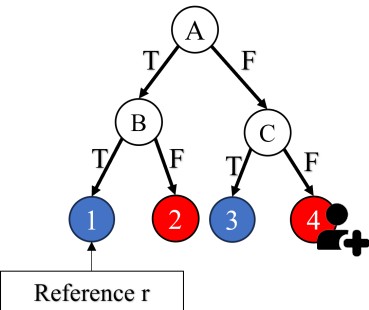

Figure 12: An counterexample with fewer reference point

Lastly, we would like to show that with all the negative leaf nodes considered as reference points if an new reference points is added, the $SEV^-$ cannot be further minimized. Since we know that the reference points should be predicted as negative, so the newly aded reference should still belongs to one of the existing negative predicted leaf node, so $SEV^-$ cannot be further minimized.

To sum up, we have proved that with the set of all negatively predicted leaves as reference points, both $SEV^-$ and the $\ell_0$ distance (edit distance) between the query and the SEV explanation is minimized.

# N    Some extra examples for different kinds of SEV metrics

Table 14: Different SEV Variants Explanations in MIMIC datasets

| | PREICULOS | GCS | HEARTRATE_MAX | MEANBP_MIN | RESPRATE_MIN | TEMPC_MIN | URINEOUTPUT |
|---|---|---|---|---|---|---|---|
| Query | 43806.28 | 10.00 | 91.00 | 29.00 | 9.00 | 34.50 | 162.98 |
| $SEV^1$ | 2215.88 | — | — | — | — | — | — |
| $SEV^F$ | 2215.88 | — | — | — | — | — | — |
| $SEV^©$ | 8739.30 | — | — | — | — | — | — |
| $SEV^T$ | — | — | — | — | — | — | 595.48 |
| Query | 0.51 | 15.00 | 105.00 | 21.00 | 20.00 | 32.28 | 7.98 |
| $SEV^1$ | — | — | — | 59.35 | — | — | — |
| $SEV^F$ | — | — | — | 59.35 | — | — | — |
| $SEV^©$ | — | — | — | 56.95 | — | 36.11 | — |
| $SEV^T$ | — | — | — | — | — | — | 595.48 |
| Query | 1.34 | 3.00 | 139.00 | 33.00 | 11.00 | 35.56 | 247.98 |
| $SEV^1$ | — | 13.89 | — | — | — | — | — |
| $SEV^F$ | — | 13.89 | — | — | — | — | — |
| $SEV^©$ | — | 9.24 | 105.96 | 59.24 | — | — | — |
| $SEV^T$ | — | — | — | — | — | — | 595.48 |
| Query | 1.64 | 11.00 | 199.00 | 14.00 | 22.00 | 37.06 | 387.98 |
| $SEV^1$ | — | — | 102.57 | — | — | — | — |
| $SEV^F$ | — | — | 102.57 | — | — | — | — |
| $SEV^©$ | — | — | 107.58 | — | — | — | — |
| $SEV^T$ | — | — | — | — | — | — | 595.48 |
| Query | 6621.40 | 13.00 | 134.00 | 28.00 | 28.00 | 34.72 | 4.98 |
| $SEV^1$ | — | — | 102.57 | — | 12.22 | — | — |
| $SEV^F$ | — | — | 102.57 | — | 12.22 | — | — |
| $SEV^©$ | — | — | 97.70 | — | 12.68 | — | — |
| $SEV^T$ | — | — | — | — | — | — | 595.48 |

Table 15: Different SEV Variants Explanations in COMPAS datasets

| | AGE | JUV_FEL_COUNT | JUV_MISD_COUNT | JUVENILE_CRIMES | PRIORS_COUNT |
|---|---|---|---|---|---|
| Query | 50.00 | 0.00 | 0.00 | 0.00 | 11.00 |
| $SEV^1$ | —- | —— | —- | —- | 2.21 |
| $SEV^F$ | —- | —— | —— | —— | 2.21 |
| $SEV^©$ | —- | —— | —— | —— | 4.63 |
| $SEV^T$ | —- | —— | —- | —— | 2.50 |
| Query | 23.00 | 1.00 | 0.00 | 1.00 | 5.00 |
| $SEV^1$ | 36.71 | —— | —- | —— | 2.21 |
| $SEV^F$ | 36.71 | —— | —— | —— | 2.21 |
| $SEV^©$ | 26.69 | 0.11 | 0.18 | 0.54 | 2.13 |
| $SEV^T$ | —— | —— | —- | —— | 2.50 |
| Query | 21.00 | 0.00 | 2.00 | 3.00 | 3.00 |
| $SEV^1$ | —- | —— | —- | 0.12 | —— |
| $SEV^F$ | —— | —— | —— | 0.12 | —— |
| $SEV^©$ | 26.69 | —— | —— | 0.54 | —— |
| $SEV^T$ | 33.50 | —— | —- | —— | —— |
| Query | 23.00 | 0.00 | 1.00 | 1.00 | 4.00 |
| $SEV^1$ | 36.71 | —— | —- | —— | —— |
| $SEV^F$ | 36.71 | —— | —- | —— | —— |
| $SEV^©$ | 26.69 | —— | —- | —— | 2.13 |
| $SEV^T$ | 23.00 | —— | —- | —— | 2.50 |
| Query | 21.00 | 0.00 | 0.00 | 0.00 | 1.00 |
| $SEV^1$ | 36.71 | —— | —- | —— | —— |
| $SEV^F$ | 36.71 | —— | —- | —— | —— |
| $SEV^©$ | 28.02 | —— | —- | —— | —— |
| $SEV^T$ | 22.50 | —— | —- | —— | —— |

Table 16: Different SEV Variants Explanations in FICO datasets

| | External RiskEstimate | MSince Oldest TradeOpen | MSince MostRecent TradeOpen | Average MInFile | Num Satisfactory Trades | NumTrades 60Ever2 DerogPubRec | NumTrades 90Ever2 DerogPubRec | MaxDelq2 PublicRec Last12M | NumInq Last6M | NumInq Last6 MExcl7days | NetFraction Revolving Burden |
|---|---|---|---|---|---|---|---|---|---|---|---|
| Query | 60.00 | Missing | 8.00 | 88.00 | 55.00 | 0.00 | 0.00 | 4.00 | 1.00 | 1.00 | 54.00 |
| $SEV^I$ | 72.21 | — | — | — | — | — | — | — | — | — | — |
| $SEV^F$ | 72.21 | — | — | — | — | — | — | — | — | — | — |
| $SEV^©$ | 70.82 | — | — | — | — | — | — | — | — | — | — |
| $SEV^T$ | 74.50 | — | — | — | — | — | — | — | — | — | — |
| Query | 60.00 | 150.99 | 32.00 | 79.00 | 8.00 | 2.00 | 0.00 | 3.00 | 0.00 | 0.00 | 112.01 |
| $SEV^I$ | 72.21 | — | 9.20 | — | 21.10 | — | — | — | — | — | 22.26 |
| $SEV^F$ | — | — | — | — | — | Missing | — | — | — | — | 9.00 |
| $SEV^©$ | — | — | 11.80 | — | — | Missing | — | — | — | — | 8.85 |
| $SEV^T$ | 74.50 | — | — | — | — | — | — | — | — | — | — |
| Query | 60.00 | 197.00 | 17.00 | 81.00 | 16.00 | 1.00 | 1.00 | 0.00 | 0.00 | 0.00 | 6.00 |
| $SEV^I$ | 72.21 | — | — | — | — | — | — | — | — | — | — |
| $SEV^F$ | 72.21 | — | — | — | — | — | — | — | — | — | — |
| $SEV^©$ | — | — | — | — | — | Missing | — | — | — | — | — |
| $SEV^T$ | 74.50 | — | — | — | — | — | — | — | — | — | — |
| Query | 59.00 | 125.99 | 12.00 | 58.00 | 18.00 | 2.00 | 1.00 | 2.00 | 10.00 | 10.00 | 95.01 |
| $SEV^I$ | 72.21 | — | — | 82.32 | — | 0.00 | — | 5.36 | 0.60 | 0.56 | 22.26 |
| $SEV^F$ | — | — | — | — | — | Missing | Missing | — | — | — | 9.00 |
| $SEV^©$ | 70.82 | 218.29 | 8.60 | 85.80 | 23.67 | 0.82 | 0.51 | 5.10 | 1.22 | 1.18 | 30.36 |
| $SEV^T$ | 74.50 | — | — | — | — | — | — | — | — | — | — |
| Query | 69.00 | 280.01 | 11.00 | 125.00 | 16.00 | 1.00 | 1.00 | 0.00 | 0.00 | 0.00 | 45.00 |
| $SEV^I$ | — | — | — | — | — | — | — | 5.36 | — | — | — |
| $SEV^F$ | — | — | — | — | — | — | — | 5.36 | — | — | — |
| $SEV^©$ | — | — | — | — | — | — | — | 5.10 | — | — | — |
| $SEV^T$ | 74.50 | — | — | — | — | — | — | — | — | — | — |

