# OpenReview forum: "Improving Decision Sparsity"
_NeurIPS.cc/2024/Conference — NeurIPS 2024 poster_

### Official Review · Reviewer_xH4u · 2024-06-28

**Soundness:** 4
**Presentation:** 3
**Contribution:** 3
**Rating:** 7
**Confidence:** 4

**Summary:**

The paper extends the notion of decision sparsity called the sparse explanation value (SEV). Cluster-based and tree-based SEV are introduced, as well as some algorithms to optimise the decision sparsity are considered. The core of the paper -- SEV -- is defined as the number of factors that need to be changed to a reference values in order to change the decision.

**Strengths:**

The problem considered is important for real (e.g., medical, criminology applications). The example (Table 2) is very helpful to understand the problem. The approach is mathematically sound and in general well-described.

**Weaknesses:**

The method is promising but its current version is hardly scalable.

**Questions:**

Line 85: "humans have no intuition for why a point belongs to one class or the other". I cannot completely agree with the statement, and it would be helpful to provide some examples. For me, on the contrary, medical doctors have often an intuition how to classify a patient, however, they do not always know how to explain their intuition.

How the sparse explanation for the sample x (mentioned on line 171) is formally defined? Is the number of features of x (original) and x (sparse) different?

**Limitations:**

The method is not scalable.

---

> ### Author Rebuttal · Authors · 2024-08-01
>
> Thank you so much for your review! We really appreciate it! See below for our response to your questions and concerns.
>
> > Line 85: "humans have no intuition for why a point belongs to one class or the other". I cannot completely agree with the statement, and it would be helpful to provide some examples. For me, on the contrary, medical doctors often have an intuition how to classify a patient, however, they do not always know how to explain their intuition.
>
> There’s a good example from Figure 3 in the paper of Keane [1] where he shows images on either side of the decision boundary that look visibly identical (we have provided the figure in the attached pdf file). There’s just no intuition why the point belongs to one class or another. We agree with you that for some decisions that, particularly where there are discrete variables, the decision boundary could be clearer. We’ll adjust the wording of that sentence. However, even for medical decisions by doctors, it’s not also clear how to diagnose someone, which is why AI can be helpful. (We have multiple projects on exactly this topic.)
>
> [1] Delaney, Eoin, et al. "Counterfactual explanations for misclassified images: How human and machine explanations differ." Artificial Intelligence 324 (2023): 103995.
>
> > How the sparse explanation for the sample x (mentioned on line 171) is formally defined? Is the number of features of x (original) and x (sparse) different?
>
> The number of features of the original query x and the sparse explanation x are the same. The sparse explanation for a specific query is defined as a p-dimensional vector that has changed the smallest number of features from its query value to the reference value to flip the prediction, while the other features remain unchanged. Four  examples are shown in Table 1. The explanations (lower rows) are the same size as the Query (top row). The gray colored values are the unchanged query values, while the black ones are changed. This is the same type of definition as other counterfactual explanation methods.
>
> > The current version is hardly scalable.
>
> SEV measures decision sparsity, so it is calculated for each point separately, therefore in practice, when we provide an SEV for one loan applicant at a time, the calculation is very fast for that applicant (fraction of a second to a few seconds, more detailed calculation time is shown in Table 7-9 in Appendix H). If you want to calculate SEV for all points at once for a huge dataset, like in the experiments of our paper, it can be computed in parallel for each point.
>
> If you want to compute it for each point and for a large number of features, we could apply the gradient-based method from our paper first in order to reduce the mean SEV of the model to be close to 1 without sacrificing the model performance. This could speed up SEV calculation for all points.
>
> Thank you so much once again for your review!

---

> > ### Comment · Reviewer_xH4u · 2024-08-07
> >
> > I acknowledge the rebuttal. Thank you for the detailed answer.

---

### Official Review · Reviewer_bvrF · 2024-07-11

**Soundness:** 3
**Presentation:** 3
**Contribution:** 2
**Rating:** 5
**Confidence:** 4

**Summary:**

The authors build on top of the Sparse Explanation Value approach by
Sun et al and provide improvements in terms of closeness and
credibility.

**Strengths:**

Sensible problem, well presented solution.

**Weaknesses:**

The main limitation I can see in the work is its very incremental nature with respect to the approach by Syn et al.: going from one negative reference point to a set by clustering negatives (cluster-based SEV) is a trivial extension, while tree-based SEV only works if the underlying model is (or can be approximated as) a decision tree, substantially restricting the applicability of the approach.

Additionally, the superiority of sparsity with respect to distance in
terms of acceptability for humans is intuitive but not always
guaranteed. When dealing with recourse, for instance, which is the
setting used in the experiments, the main problem is the cost of the
change, and slightly modifying two features could be less
expensive than modifying a single one by a larger value. This should
be better discussed in the paper.

**Questions:**

How can you turn a DT leaf into a reference point when dealing with
continuous features? what is the actual value of the continuous
variables in the leaf?

**Limitations:**

The approach is very incremental, and the main extension substantially
restricts the applicability of the method.

The rebuttal of the authors did shade additional light on the novelty of the contribution.

---

> ### Author Rebuttal · Authors · 2024-08-01
>
> Thank you so much for your review! We really appreciate it! See below for our response to your questions and concerns.
>
> > The work is its very incremental nature with respect to the approach by Sun et al. cluster-based SEV is a trivial extension, while tree-based SEV only works if the underlying model is a decision tree, substantially restricting the applicability of the approach.
>
> In terms of your question about the contribution of our paper, we understand that some of the ideas seem trivial (going from one negative reference to a set), and that you think the extension to decision trees is not broadly useful because it applies only to trees. Indeed the idea of going from one reference to many is easy. Working out how to do it, and to add credibility to the computation, is not. In terms of trees, there are many papers at NeurIPS each year that are *only* focused on decision trees! Trees are some of the most popular algorithms for interpretable machine learning, since the 1960’s. The fact that we have a substantially better way to define and compute SEV for trees is important for a lot of applications! Previous studies have also taken tree-based models for separate discussion. For instance, there exists a lot of papers discussing about fast shap values computation in tree-based models, also known as treeSHAP.
>
> While it’s tempting to believe ideas are incremental in retrospect, they are often not obvious until after seeing them. Many of our ideas are not obvious at all, for instance looping through all good sparse decision trees to find one that optimizes SEV + accuracy, which makes what would have been a practically impossible computation now take seconds. Many important ideas in ML seem obvious in retrospect, e.g., placing skip connections in neural networks. If you glance through the orals in NeurIPS from last year, they are almost all variations on existing topics. Our paper is only the second paper on SEV, which is a new notion of sparsity, and it generalizes the definition and works out how to optimize it and make its computations credible.
>
> We have actually made a lot of improvements and generalizations with respect to the original SEV paper after generalizing the framework of SEV to make it support instance-wise reference selection.
> - We proposed cluster-based SEV and its variants in order to handle the two objectives of instance-wise selection: the flexible reference solves the issue of higher SEVs (cluster SEVs are larger than standard SEVs), and the credibility approach enables the explanations to be located in a high density region. The credibility setup is definitely not obvious.
> - For tree-based models, we introduced SEV-T which directly uses negative leaf nodes as the reference points, and introduced a fast calculation procedure for SEV-. That fast procedure is definitely not obvious and is comparable to content within an algorithms course. The bound on credibility in SEV- in Theorem 4.3 isn’t obvious either - tree-based SEV- computations are always sufficiently credible.
> - We generalized the original gradient-based methods and introduced new search-based methods specifically for tree-based models in order to optimize the specific model type to have low SEV. The methods for optimizing SEV are not obvious, particularly Sec 5.2 where we optimize SEV by scanning all good trees using the new Rashomon set algorithms and finding the one with the best SEV. (In contrast, think about how many papers optimize some combination of $ell_1$ loss with accuracy, or optimize neural networks using some variation of gradient descent!)
>
> > The superiority of sparsity with respect to distance in terms of acceptability for humans is intuitive but not always guaranteed. When dealing with recourse, for instance, slightly modifying two features could be less expensive than modifying a single one by a larger value.
>
> Remember that SEV is not recourse, it is a measure of sparsity only. That said, it’s possible to change the metric to a recourse metric if desired. We did use multiple distance metrics. In our experiments we also used the $L_\infty$ metric to evaluate the cost of change, which would handle the case you mentioned - when we care about how much we change the features. Table 6 in Appendix G provides the mean $L_\infty$ for each explanation. Based on Table 6, SEV-C achieves better sparsity, relatively low $L_\infty$ change (compared to DiCE under the same sparsity level), and meaningfulness (high median log likelihood) *simultaneously,* while for the other methods except DiCE, they use almost all features in their explanations, which is not only too complex to interpret, but also is not actionable in practice, and would yield worse recourse scores. This is not the case of modifying two features instead of one: it is modifying all instead of one. So already we would be better in terms of most recourse loss functions. It is possible to specialize the computation to a particular recourse loss, but that’s not our goal here, we’re concerned with sparsity of the explanation, not other costs. We will put a note for future work by others.
>
> > How can you turn a DT leaf into a reference point when dealing with continuous features? What is the actual value of the continuous variables in the leaf?
>
> Interestingly, the reference can be any point x within the leaf. So if the leaf is defined by $x_1>5$ and $x_3>0$, then any point with those conditions is a viable reference. For a query, the algorithm needs to flip its feature values to satisfy those of the leaf conditions to make an opposite prediction.
>
> Since you can choose any point in the leaf as a reference value, you could choose the median/mean values of points in the leaf. That choice won’t influence the fast calculation of SEV-T, so the user can choose any actual value inside the leaf that they believe is most meaningful as the leaf’s reference.
>
> Thank you so much once again for your review!

---

> > ### Comment · Reviewer_bvrF · 2024-08-12
> > **Thanks for your answer**
> >
> > Thanks for your detailed feedback, I appreciate the clarifications and your arguments in favour of the novelty of the contribution, I encourage you to better clarify these aspects in the manuscript. This said, I am happy to raise my score to a borderline accept.

---

> > > ### Author Response · Authors · 2024-08-12
> > >
> > > Thank you for the insightful comments and willingness to raise your score! We will add those clarifications to the revised manuscript.

---

> ### Author Response · Authors · 2024-08-12
>
> Dear reviewer brvf,
>
> As the discussion period is approaching its conclusion in two days, we would like to kindly remind you that we have addressed your comments in our rebuttal. We would greatly appreciate any additional feedback you may have before the deadline. If you have any further questions or concerns, please do not hesitate to reach out, and we will do our utmost to respond promptly.
>
> Thank you for your time and consideration.
>
> Best regards,
>
> The Authors of Submission 11304

---

### Official Review · Reviewer_PbJJ · 2024-07-12

**Soundness:** 4
**Presentation:** 4
**Contribution:** 4
**Rating:** 8
**Confidence:** 2

**Summary:**

This paper proposes several ways to create closer, sparser, and more credible explanations for the SEV, along with two optimizing models. The results of the experiments on various datasets support the paper's claims.

**Strengths:**

1. Before reading this paper, I was unfamiliar with the sparse decision field. However, this paper is well-written and enjoyable to read.
2. I think this paper is quite creative by simultaneously considering closeness, sparsity, and credibility.
3. The comprehensive experiments address most of the claims they proposed in the introduction section.

**Weaknesses:**

It could be better to provide the complexity analysis and the analysis of time expenditure for different variants of SEV. For example, the computational benefits of using tree-based SEV.

**Questions:**

1. In Table 1, what do the gray numbers represent?
2. In line 199, there are duplicate "and"s.

**Limitations:**

I'm concerned about the depth of the tree if the model applies to large-scale data. In section A, the number of observations is up to 100K. If we encounter a much larger data set, the complexity of the model would exponentially increase with the depth of the tree. That's why I'm interested in time analysis.

---

> ### Author Rebuttal · Authors · 2024-08-01
>
> Thank you so much for your review! We really appreciate it! See below for our response to your questions.
>
> > In Table 1, what do the gray numbers represent?
>
> Thank you for pointing out the missing explanation for the gray numbers. The gray numbers in Table 1 represent the query feature values that haven’t been changed, while the black numbers are the sparse explanations that have been made in order to make the prediction flipped. You can observe that the gray feature values are the same as the query feature values. In SEV, we can only use the changed features as the explanations, which is the same approach as DiCE.
>
> > In line 199, there are duplicate "and"s.
>
> Thanks for pointing out the typo! We have already corrected the typo in our paper.
>
> > It could be better to provide the complexity analysis and the analysis of time expenditure for different variants of SEV. For example, the computational benefits of using tree-based SEV. I'm concerned about the depth of the tree if the model applies to large-scale data. In section A, the number of observations is up to 100K. If we encounter a much larger data set, the complexity of the model would exponentially increase with the depth of the tree. That's why I'm interested in time analysis.
>
> Thanks for your question about the time complexity for tree-based SEV. The time complexity of tree-based SEV calculation is based on the structure of the tree instead of simply the depth of the tree. The worst case for an tree-based SEV calculation is $O(n)$, where n is the number of nodes in the decision tree. The worst case does not scale as the number of data points. Moreover, $\text{SEV}^T$ will be stopped early when the query has a sibling leaf node with negative prediction (shown in theorem 4.1), or the query already finds an explanation with $\text{SEV}^T$ = 1 during the search (see Line 28-29 in Algorithm 3). Therefore, the calculation of $\text{SEV}^T$ remains fast with the growth of observations.
>
> Thank you so much once again for your review!

---

> > ### Comment · Reviewer_PbJJ · 2024-08-09
> >
> > Thanks for your answers!

---

### Author Rebuttal · Authors · 2024-08-01

Thank you to all reviewers. The following pdf goes with the response for Reviewer xH4u.

---

### Comment · Area_Chair_XV6A · 2024-08-09
**Discussion**

Hi all, the authors have provided detailed responses to the reviews. Please read the responses, and take advantage of the reviewer-author discussion period to ask for additional clarifications if needed.

In particular, reviewer bvrF has given the lowest score due to a number of issues she/he identified with the submission. The authors addressed these issues in their response. Does the reviewer agree with the response?

---

### Decision · Program_Chairs · 2024-09-25

**Decision:**

Accept (poster)

**Comment:**

The authors present enhancements on the Sparse Explanation Value (SEV) framework, proposing cluster-based and tree-based variants that improve the explanations provided.  The contributions of this work are significant from a technical and practical point of view.  After the author feedback, all reviewers recommended accepting the paper.